# Analysis of Acoustic Emissions for Determination of the Mechanical Effects of Scratch Tests

**Timothy Devenport [1], Bernard Rolfe [2] , Michael Pereira [2] and James M. Griffin [1],\***

1   Institute for Clean Growth and Future Mobility, Coventry University, Coventry CV1 5FB, UK; devenpot@uni.coventry.ac.uk
2   School of Engineering, Deakin University, Geelong, VIC 3216, Australia; bernard.rolfe@deakin.edu.au (B.R.); michael.pereira@deakin.edu.au (M.P.)
\*   Correspondence: ac0393@coventry.ac.uk

**Featured Application: In this paper, the applicability of acoustic emission (AE) measurement of galling wear in sheet metal forming (SMF) is investigated.**

**Abstract:** Acoustic Emission (AE) is a promising technique for measuring tool wear online and in real time. In this work, scratch tests were conducted to better understand the "pre-wear" AE response based on loading conditions that were not sufficient to generate galling. The scratch tests used the same type of indenter against two different sheet materials: aluminum and steel. The results showed that AE parameters such as the mean frequency, Centroid frequency and Shannon entropy outperformed other frequency domain techniques by discriminating between the two sheet materials in scratch tests. From the literature, the frequency region of interest was expected to be sub 300 kHz. However, in this study, activity below this threshold was found to be noise, whereas distinct frequencies were found at much higher frequencies than expected. These results are compared against single grit "SG" tests of both mild steel- and nickel-based superalloys to allow comparison of the two test methods and materials used. This comparison showed that the SG tests excited the acoustic emission in ways in which the scratch tests did not. Another factor when using acoustic emissions to monitor sheet metal forming is the differences obtained in energy–frequency mapping, where many report the galling phenomena between a certain amplitude and frequency range. Such results are specific to the setup and the materials/geometries used. Further work presented here compares different scratch tests where energy–frequency mapping is different for different materials/geometries.

**Keywords:** acoustic emission; scratch tests; mechanical inspection tests; force (load); wear mechanisms

## 1. Introduction

Sheet metal forming (SMF) is a common manufacturing technique used to make components such as automotive body panels and aircraft skins. It is widely utilized in industry for high-volume production. SMF is a fast, cheap, near net shape (limited scrappage) process which produces high-quality parts when galling does not occur. The galling damage mechanism is of high industrial relevance, particularly for the SMF industry, where restrictions on lubricants and the use of high-strength sheet materials may lead to galling [1]. Galling wear of forming tools is a well-known problem in SMF and can result in a combination of adhesive and abrasive wear [2]. During the galling wear process, material is transferred from the sheet to the tool, thus damaging the tool and future parts [3]. The definition according to ASTM G40-17 is "a form of surface damage arising between sliding solids, distinguished by macroscopic, usually localized, roughening, and the creation of protrusions above the original surface; it is characterized by plastic flow and may involve material transfer" [4]. The most important factor affecting the quality of the final products

in SMF processes is wear control in the contact area between the sheet material and the tool surface [5]. Galling failure accounts for up to 71% of the cost for die maintenance [6].

It is widely reported that galling exhibits three distinct regimes described by an increasing coefficient of friction [7–9]. Initially, contact is between a clean tool and the sheet surface. However, continued sliding leads to an intermediate stage through local adhesive wear, and this gives rise to individual lumps of adhered sheet material on the tool surface. At this point, the (local) contact becomes a sheet-to-sheet material contact. Further sliding in this first regime grows the adhered lumps to a final stage of microscopic scratching. In the second regime, adhesive and abrasive wear mechanisms interact, and growth of lumps occurs primarily where sheet-to-sheet contact prevails. In the third regime, the entire contact area becomes covered by a layer of sheet material, which leads to severe adhesive wear of the entire sheet track [10]. It has been established by multiple other works that these three regimes are detectable by acoustic emissions (AE) under many different experimental regimes [11,12].

AE is showing promise as a method for online tool condition monitoring by measuring elastic waveforms in the forming tools during manufacture. AE is the release of transient elastic energy accumulated in a material during deformation processes. Skåre et al. identified that the main drawback of AE as a method of measuring friction processes or initial cracking in a forming process is that several phenomena can appear as the same signal when analyzed in the time–frequency domain [13].

Behrens et al. showed that defective products could be detected by AE during manufacturing by analyzing simple signal parameters; however, the superposition of a signal from a defective component and one from tool wear could not be distinguished when analyzing the AE in terms of amplitude and energy [14]. Behrens et al. did not investigate if time–frequency techniques could separate this super-positioning. Wang and Wood found that a strong relationship between the AE root mean square (RMS) and the coefficient of friction (COF) exists under steady state using cross-correlation analysis (based on instantaneous features in both AE RMS and COF) [15]. A key conclusion of Wang and Wood is that the AE energy rise due to failure is a characteristic of the ball/disk material combination. Shanbhag et al. also proposed the use of mean frequency to infer the progression of galling wear and showed that decreasing mean frequency was indicative of increasing wear [16]. Hase et al. also showed that materials are separable during wear by both counts per mm and AE pulse energy [17].

Moghadam et al. studied the AE of galling from a metal-forming point of view by using the bending under tension test. They used strips of 304 L stainless steel formed over a powdered metallurgy (PM) tool steel and two different lubricants and found the AE signature for galling to be 20 to 160 kHz [18]. Hase et al. conducted research to distinguish between abrasive and adhesive wear mechanisms using AE monitoring techniques. Hase et al. reported the frequency peak of the AE signals occurs at around 1100 kHz for adhesive wear, whilst the frequency peaks are distributed in the region from 250 kHz to 1000 kHz for abrasive wear; thus, wear mechanisms can be recognized from the features of the AE frequency spectrum. Shanbhag et al. investigated wear in stamping and showed that adhesive wear ranged from 100 to 500 kHz, and abrasive wear from 100 to 200 kHz [19]. Chen et al. and Griffin investigated the AE response of different aerospace materials (NI-based alloys and steel alloys) when undergoing single grit scratch tests [20,21]. By analyzing the AE signal with Short-time Fourier Transform (STFT) methods, Chen et al. and Griffin showed that each of the four materials resulted in distinct frequencies from 100 kHz to 575 kHz. This result contradicts the findings of Hase et al., which state that all materials should behave similarly. For the SG tests, the prominent AE feature frequencies of the scratches are expected to fall in the range 100~550 kHz, which are similar to the AE feature frequencies in grinding tests experienced in previous work.

Sindi et al. correlated the acoustic emission to the deformation mechanisms associated with the damage mechanisms during SMF [22]. Their key results showed the specific

damage mechanism in relation to the amplitude and duration of the waves and frequency content (see Figure 1).

| | Wave characteristics | Damage mechanism (Galling) | Micrograph | Dominant frequency range (kHz) |
|---|---|---|---|---|
| (a) | Amp. ≤ 0.01 (mV) Dur. < 0.5 (ms) | Plastic deformation (Stage I) | | 60—125 |
| (b) | 0.01 ≤ Amp. ≤ 0.05 (mV) 0.5 ≤ Dur. ≤ 1.0 (ms) | Material transform & Severe scratching (Stages II & III) | | 199—375 |
| (c) | Amp. > 0.05 (mV) Dur. > 1.0 (ms) | Adhesive wear & rupture of junction (Stage III) | | 250—310 |

**Figure 1.** Wave characteristics for the three damage mechanisms and Dominant frequency ranges of the AE signals emitted during the three damage mechanisms. (**a**) stage 1 galling wear. (**b**) stage 2 and 3 galling wear. (**c**) stage 3 galling wear Reprinted/adapted with permission from [22].

According to these works, the frequency response for sliding wear galling could be anywhere between 60 kHz and 1100 kHz, depending on the wear mechanism and/or test set up. However, it is expected that the initial signs of galling wear result in signals at the lower end of this range.

None of these works report on the AE response during "normal" or "pre-galling" operating conditions, which will be key for developing the AE technology and techniques for detecting and reacting to the development of galling wear. Therefore, our research uses scratch testing to investigate the AE response of different materials when the applied load is insufficient to produce galling, utilizing Fourier analysis, mean frequency, centroid frequency, and Shannon entropy. The results show that the AE frequencies and behavior are specific to the setup and the materials/geometries used.

## 2. Experimental Methods

In this work, scratch tests were conducted to better understand the "pre-wear" AE response and based on loading conditions that were not sufficient to generate galling. Scratch testing has been used to investigate the AE response during galling wear for metal forming applications in other works in the literature [16,19,23,24]. These results are compared against single grit "SG" tests on nickel-based superalloys to allow comparison of the two test methods and materials used at loads high enough to produce galling. SG testing has been used to display the differences of AE when grit and different materials interact. This is important, as AE ranges can differ between different material phenomena interactions.

### 2.1. Scratch Tests

Scratch tests were performed to compare the differences in AE from scratching differing materials under differing loading conditions. The scratch tests were performed using

a Bruker TriboLab machine, Figure 2. The parameters for each test are shown in Table 1. Samples were scratched against a 6-mm-diameter tool steel ball bearing under a constant speed (1 mm/s) and both constant (1 kN) and increasing loading profile (0.3 to 1.5 kN), for a scratch length of 20 mm and 40 mm, respectively. The setup is shown in Figure 2.

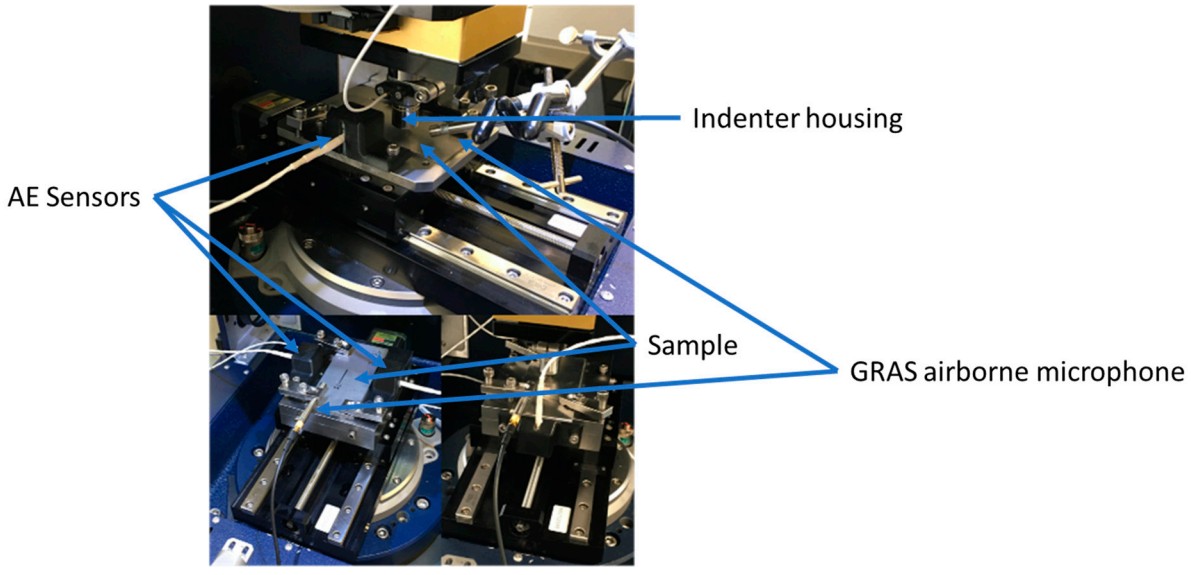

**Figure 2.** Bruker TriboLab test set up.

**Table 1.** Test Parameters/details.

| | Indenter Material | Plate Material | Loading Profile | Load (N) | Length (mm) | Duration of Scratch (s) | AE Sensor Position |
|---|---|---|---|---|---|---|---|
| Test 1 | Tool steel | Steel SA52100 | Increasing | 300–1500 | 40 | 40 | A |
| Test 2 | Tool steel | Aluminium | Increasing | 300–1500 | 40 | 40 | B |
| Test 3 | Tool steel | Aluminium | Constant | 1000 | 20 | 20 | C |
| Test 4 | Tool steel | Steel SA52100 | Constant | 1000 | 20 | 20 | C |

Post-test, all samples and indenters were investigated optically (equipment: Alicona-InfiniteFocus; objective magnification: 5×; vertical resolution: 1.4 μm; scan area: 23 × 10 mm and lateral measurement range of 0.16 mm). By using optical profilometry, the visual observations of the scratch manifestation and associated material built-up edge were scanned using the non-contact optical profilometer.

### 2.2. Single Grit Methods

The SG scratch tests were carried out on a Makino A55 Machine Centre, as shown in Figure 3. An $Al_2O_3$ single grit was glued into a microscopic drilled hole in a steel plate so that it protruded from the surface. This way, the SG was the first object to make contact with the workpiece, as shown in Figure 4, to give an average scratch depth of approximately 1 μm. The steel plate was rotated at commercial grinding speeds towards the flat horizontally placed workpiece. The workpiece material was varied to note AE differences when the single grit material remained constant for all tests. The two AE sensors were set-up at equal distances apart, Figure 3. If AE differences resulted from these different materials, then analysis to such effects will be commented on.

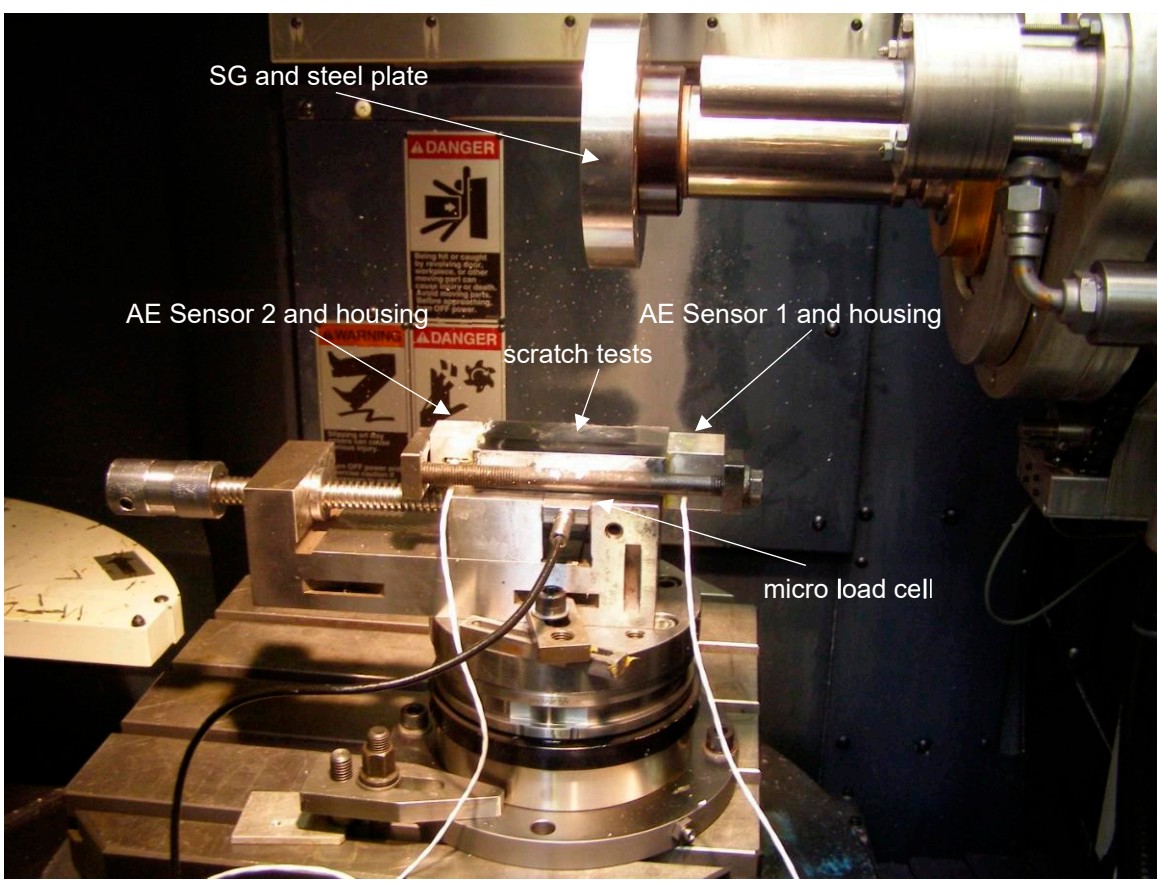

**Figure 3.** Single grit test set up. Reprinted from Griffin 2015 [21].

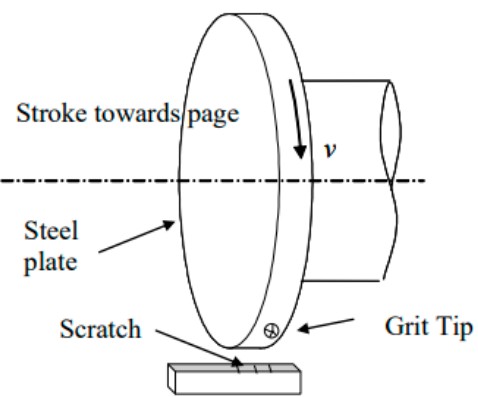

**Figure 4.** Schematic of horizontal single grit test rig. Reprinted from Griffin 2015 [21].

*2.3. Acoustic Emission Sensors and Data Processing*

The AE apparatuses used for all tests were two contact Physical Acoustics WD AE sensors connected to a data acquisition (DAQ) unit with two pre-amplifiers. The amplification to each sensor was maintained at 20 dB, with a 35 dB threshold selected for the DAQ. Based on the sensor power spectrum, AE devices are suitable for a range of frequencies between 70 kHz and 1 MHz. The positions of the sensors were moved between tests to observe the effect of the positions of the sensors on the measured AE signal. These positions were on the scratch face parallel to the scratch direction, on the scratch face perpendicular to the scratch direction, on the "edge" of the sample parallel to the scratch direction, and on the "edge" of the sample perpendicular to the scratch direction. A schematic of the sensor

positions is given in Figure 5. It must be stressed that only 2 AE sensors were used during any one test, as indicated by the sensor configuration described in Table 1.

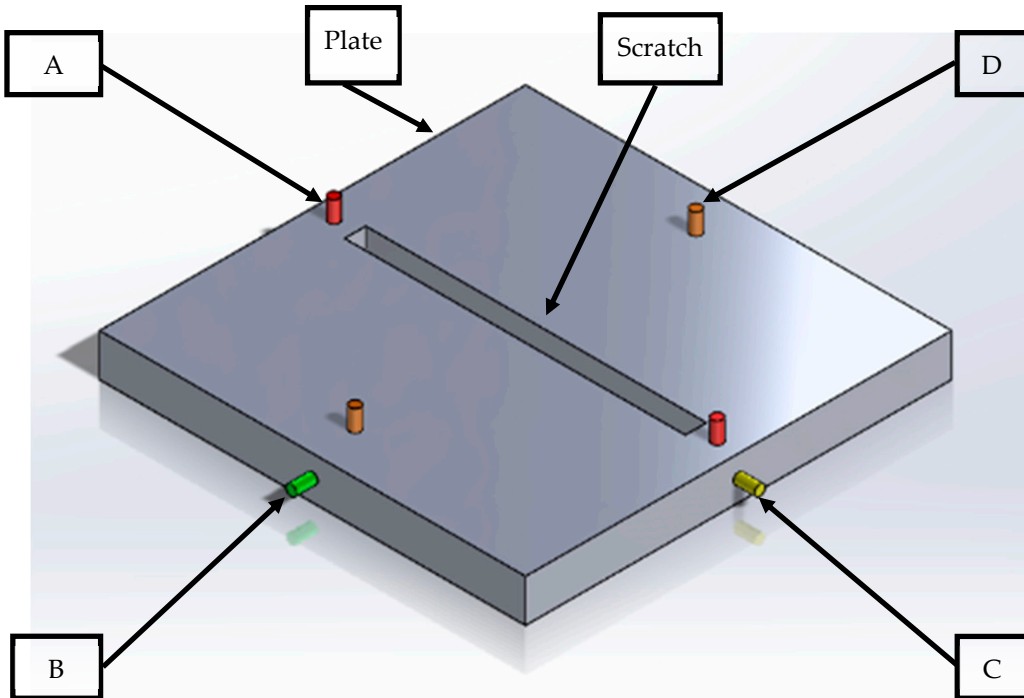

**Figure 5.** Schematic of test sensor positions. A = on the scratch face parallel to the scratch direction (red markers). B = on the "edge" of the sample perpendicular to the scratch direction (green markers). C = on the "edge" of the sample parallel to the scratch direction (yellow markers). D = on the scratch face perpendicular to the scratch direction (orange markers). Note measurements from orientation D are not used in this work.

It is known from the literature that it is critically important to select the most appropriate AE parameters [25] to minimize the probability for error. It is important to select some parameters that are a function of the peak voltage (which may be influenced by the researchers' choice of AE setup) and some that are waveform-dependent and therefore independent of the AE setup [26].

The study of the application of AE is quite advanced in the study of failure modes in fiber-reinforced plastics and structural health monitoring [25], where parameters such as frequency, amplitude, duration, rise time, peak amplitude, energy, counts, centroid frequency, weighted peak frequency, partial power, number of hits, and counts per events are commonly used, as described by Barile [26]. Only a select few of these parameters have been used in the application of AE to galling wear [11,16,17,19,22,27,28].

Therefore, in the scratch tests in this work, analysis of the AE signal was conducted using the AE parameters hits (hits were unable to be used, as the threshold was set too low, such that the background noise was enough to register a hit and therefore no useful information could be inferred), counts, and standard time–frequency techniques; fast Fourier transform (FFT) and short time Fourier transform (STFT) was calculated using MATLAB's toolboxes. Additionally, a novel technique, mean frequency (mean frequency cannot be calculated for the single grit tests, owing to the short duration of the AE burst), is compared against more established techniques, Shannon entropy and Centroid frequency. For reasons explained later, a 300 kHz high pass filter was applied to the raw AE data using MATLAB's built-in Digital Signal Processing (DSP) toolbox. Mean frequency was calculated by windowing the signal into windows of 0.5 s and calculating the mean in each window using MATLAB's meanfreq function (Mean frequency—MATLAB meanfreq—MathWorks

Australia). The sampling rate for the FFT, STFT, and mean frequency was set to 2 MHz as a compromise between preventing aliasing and capturing an unmanageably large data set.

For analysis of the SG AE, the signal was reconstructed using the MATLAB DSP Toolbox and all the short-burst high-frequency information was obtained. By using a Chebyshev Type II bandpass filter with a cut off frequency of 80 kHz to 1 MHz, most of the noise generated was eliminated. The sampling rate was set to 5 MHz to prevent aliasing.

### 2.4. Materials

For the scratch tests, two materials were used for the sheet, aluminium H8 temper and steel (SA52100); the properties are given in Tables 2 and 3. The counter face was a commercially available steel ball bearing made of tool steel.

**Table 2.** Composition of materials used in this study.

| Element (Wt. %) | Steel SA52100 (ASTM SAE AISI 52100 Steel Properties, Composition, Equivalent (theworldmaterial.com) (accessed on 23 January 2022) | Inconel 718 (Inconel 718 | Material Datasheet (inconel-718.com) (accessed on 23 January 2022) | CSMX4 (AISI CMSX-4 Nickel Alloys: Chemical Composition & Other Alloy Properties. (alloytester.com) (accessed on 23 January 2022) | EN8 Steel (Engineering Steel EN8 (080M40) (smithmetal.com) (accessed on 23 January 2022) | MARM-002 (Mar-M002 | HB SPEICAL ALLOY (hb-specialalloy.com) (accessed on 23 January 2022) |
|---|---|---|---|---|---|
| Aluminium, Al | | | 5.6 | | 5.5 |
| Boron, B | | | | | 0.015 |
| Carbon, C | 0.980–1.10 | | | 0.36–0.440 | |
| Cobalt, Co | | | 10 | | 8.25 |
| Chromium, Cr | 1.30–1.60 | 18 | 7 | | 5.5 |
| Iron, Fe | 96.5–97.32 | 18.3 | | Balance | 0.5 |
| Manganese, Mn | 0.250–0.450 | | | 0.6–0.1 | |
| Molybdenum, Mo | | 3 | 0.6 | | 0.7 |
| Nickel, Ni | | 53.7 | 67 | | 59 |
| Niobium, Nb | | 5.1 | | | |
| Phosphorous, P | <0.0250 | | | 0.04 | |
| Rhenium, Re | | | 3 | | |
| Silicon, Si | 0.150–0.300 | | | | |
| Sulphur, S | <0.0250 | | | <0.05 | |
| Tantalum, Ta | | | | | 3.0 |
| Titanium, Ti | | 0.9 | 1 | | 1.0 |
| Tungsten, W | | | 6 | | 10 |

**Table 3.** Properties of materials used in this study.

| Property | Aluminium H8 Temper | Steel SA52100 | Inconel 718 | CMSX4 | EN8 Steel | MARM-002 |
|---|---|---|---|---|---|---|
| Density (kg/m$^3$) | 2710 | 7700–8030 | 8193 | 8690 | 7800–8030 | 8267 |
| Hardness (HV) | 44 | 848 | 456 | 520 | 178 | 470 |
| Tensile Strength (MPa) | 400 | 590 | 758–1407 | 1090 | 510–660 | 965 |
| Yield Strength (MPa = N/mm$^2$) | 335 | 360 | 1150 | 1150 | 245–530 | 815 |
| Elastic Modulus (GPa) | 68.3 | 190–210 | 31 | 18.5 | 200 | 24.6 |
| Elongation (%) | | 20 | 21–27 | 10–12 | 32.8 | 13–17 |

For the SG tests, four commercially important aerospace super alloys were used, Inconel 718, CSMX4, EN8 steel, and Marm-002 (where materials one, three and four are nickel based). Such materials were chosen to give different material characteristics when exerted with a source of initiated stress. All samples of Ni-based alloys used were polished to Ra = 0.01 μm. In this case, the original experiment was designed to investigate different levels of cutting, ploughing, and rubbing, and what these physical actions mean from an acoustic energy perspective, especially when applied to different materials.

## 3. Results

In this section, the results from the increasing load and constant load scratch tests will be presented first (Sections 3.1 and 3.2). Then, the results from the SG tests will be presented (Section 3.3).

### 3.1. Scratch Tests with Increasing Load

The raw data of the scratch tests are given in Figure 6. Steel under increasing load exhibits multiple distinct AE events, whilst aluminium under increasing load does not show such distinct AE events. For these tests, the sensors were in orientation A for the steel sample (on the scratch face parallel to the scratch direction) and orientation B for the aluminium sample (on the "edge" of the sample perpendicular to the scratch direction).

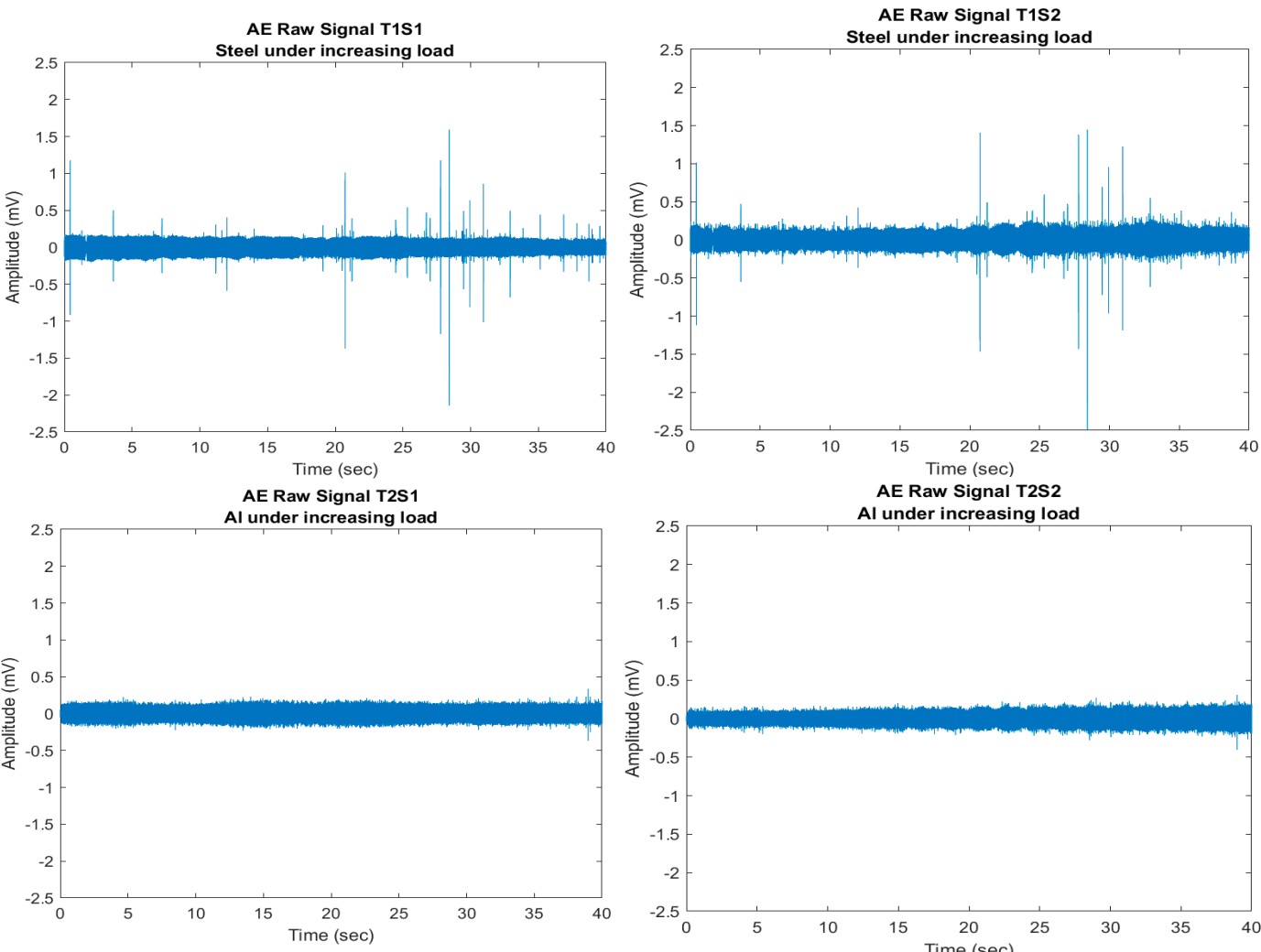

**Figure 6.** Raw signal of the scratch tests under increasing loads.

For the scratch tests, the STFT spectrogram (Figure 7) shows dominant frequencies in a low-frequency region below 300 kHz were persistent throughout the duration of all the tests and are therefore believed to be machine noise, as this is seen in all tests. Moreover, the STFT spectrogram of the steel/steel tribopair does not appear to show any significant activity above 300 kHz. However, there appears to be some activity in the steel/aluminium tribopair, which has been attributed to greater interaction between the indenter and plate materials, as evidenced by the deeper penetration depth and higher measured friction shown in Figure 11.

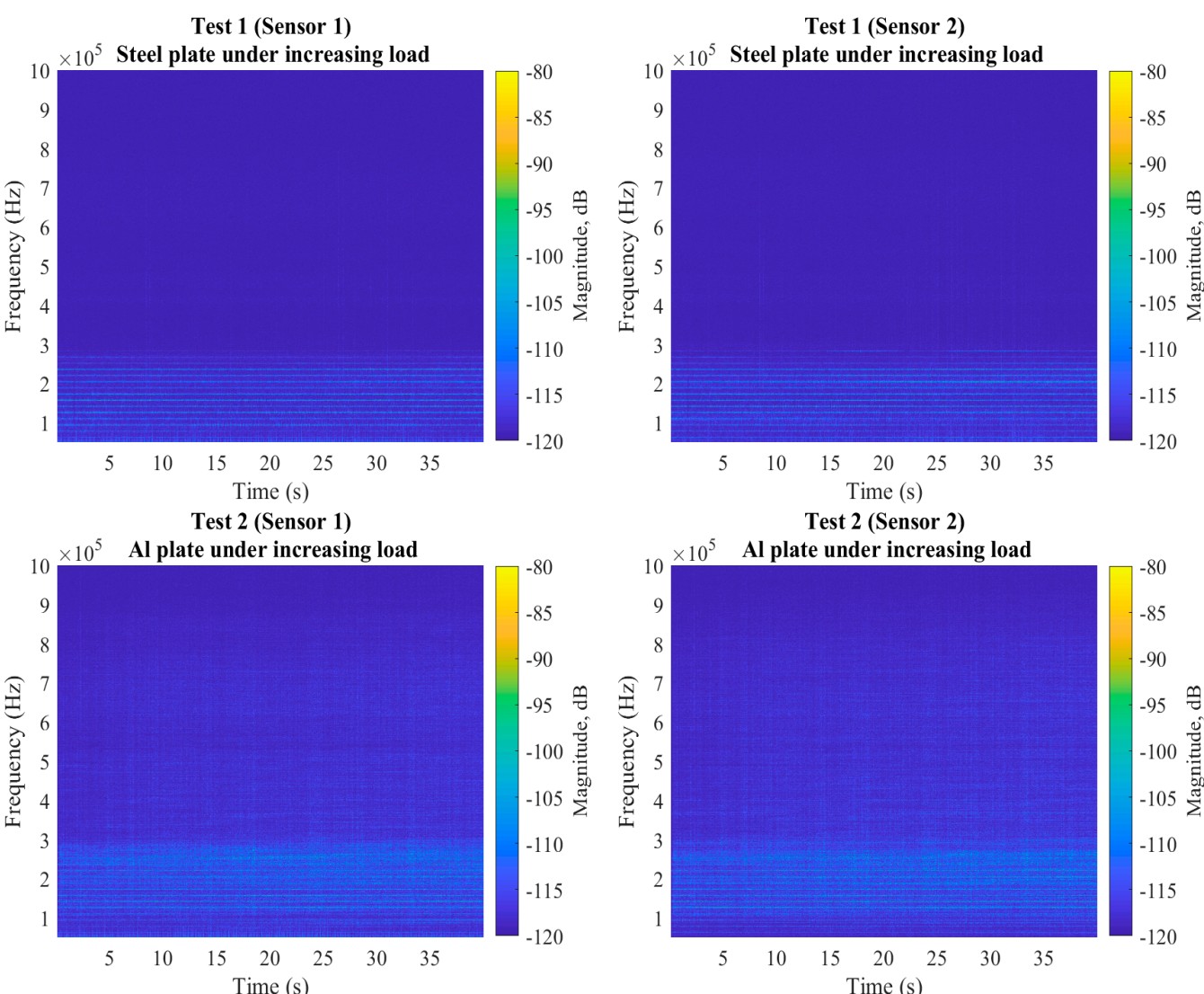

**Figure 7.** STFTs for scratch tests of tool steel indenter on steel sheet (**top**) and tool steel on aluminium sheet (**bottom**) under increasing load.

The mechanical data show the depth of cut increased linearly for both steel and aluminium, whilst the frictional force for aluminium increased slightly faster than for steel, Figure 8 (below).

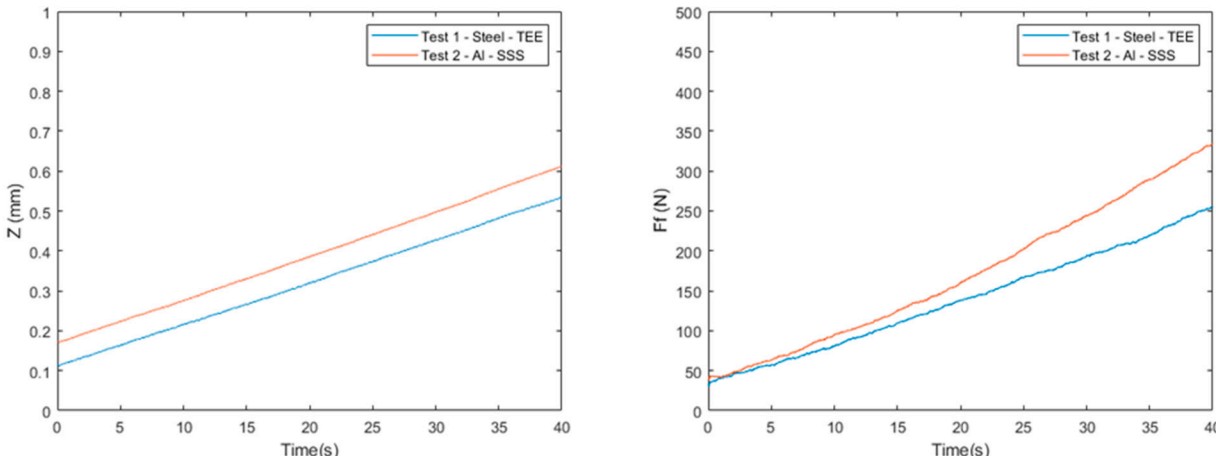

**Figure 8.** Mechanical data for tool steel indenter on tool steel sheet (test 1) and tool steel on aluminium sheet (test 2) under increasing load. Depth of cut (**left**) and friction force (**right**).

The FFT analysis shows (Figure 9) substantial activity below 300 kHz, as expected from the noise seen in the STFT analysis, as well as three dominant peaks at approximately 700 kHz, 800 kHz, and 950 kHz for both plate materials.

**Figure 9.** FFTs of tool steel indenter on steel sheet (**top**) and tool steel on aluminium sheet (**bottom**) under increasing load for scratch tests. (**left**), sensor 1. (**right**), sensor 2.

Figure 10 (left) shows that the mean frequency for the steel plate exhibits lower mean frequency than the aluminium plate with a gentle increase across the duration of the test. The aluminium also exhibits this increase in mean frequency with increasing load. Figure 10 (right) shows the mean frequency after a high pass filter at 300 kHz has been applied to the signal, which now shows the steel plate exhibiting a higher mean frequency (approximately 600 kHz) than the aluminium (500 kHz). The mean frequency for the steel appears to gently decrease with increasing load, whilst the aluminium gently increases with increasing load after this filter has been applied.

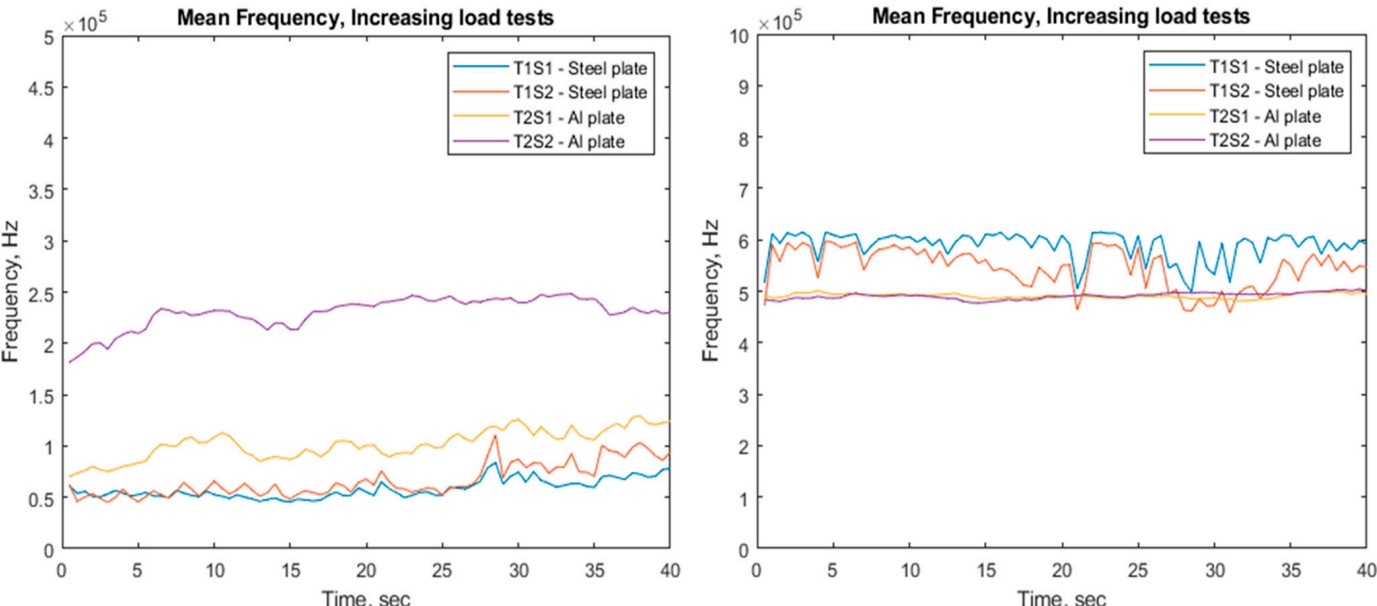

**Figure 10.** (**Left**) Mean frequency across test of tool steel indenter on tool steel sheet and tool steel on aluminium sheet under increasing load, including the sub-300 kHz region. (**Right**) Mean frequency of same increasing load tests with 300 kHz high pass filter applied.

The centroid frequency (Figure 11 top) shows the steel typically has a higher centroid frequency (600 kHz) than the aluminium (approx. 500 kHz) throughout the duration of the increasing load test. However, the centroid frequency for the steel remains almost constant, whilst the aluminium exhibits a very gentle increase in centroid frequency. However, Shannon entropy (Figure 11 bottom) shows an increase with load for the aluminium sample not seen in the steel sample.

Whilst AE hits and counts were measured, the threshold for the hits was set too low such that noise was sufficient to trigger a hit. Moreover, there was little to no correlation between the counts and the scratch, such that the period where the scratch occurred could not be found within the counts data, so these results have not been shown here.

### 3.2. Scratch Tests with Constant Load

Figure 12 shows the raw AE signal from the constant load scratch tests. Again, the steel plate exhibits several distinct AE events (as also shown in the increasing load tests), whilst the aluminium this time does show some AE events of smaller magnitude (than the steel) but more than the increasing load tests. For these tests, the sensors were in orientation C for both samples (on the "edge" of the sample parallel to the scratch direction).

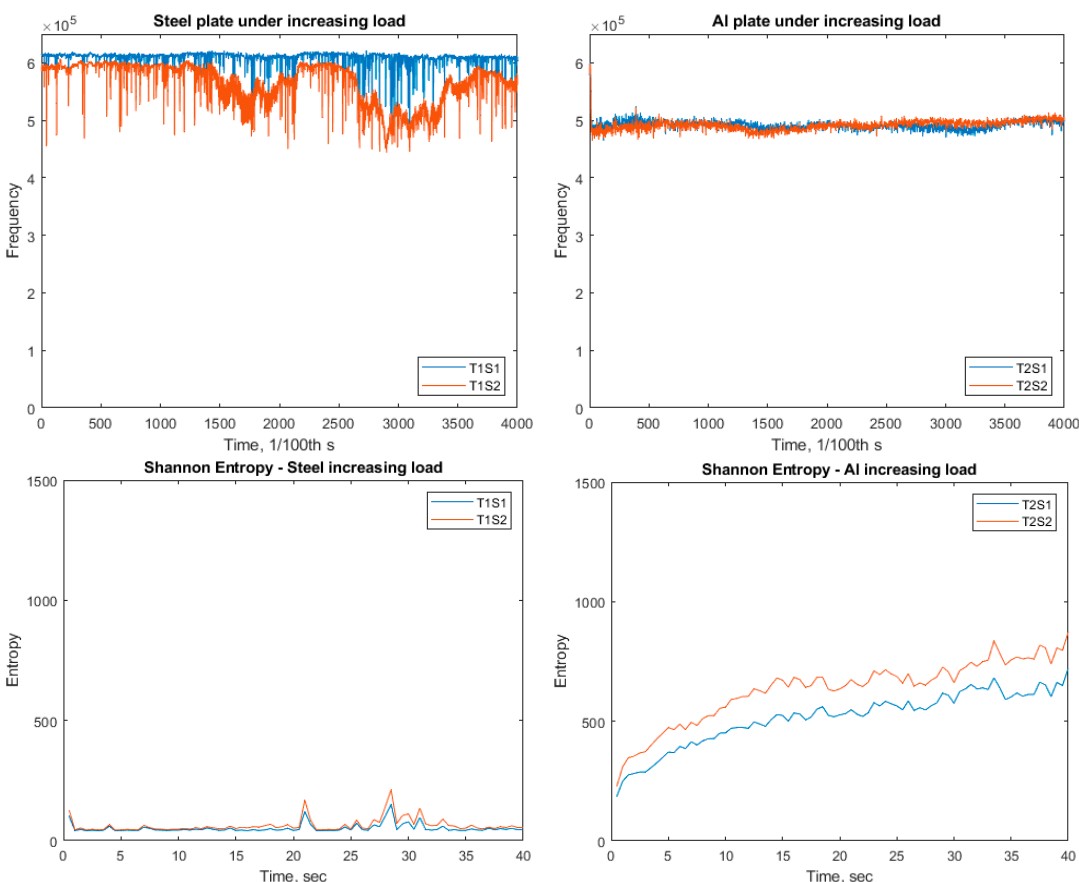

**Figure 11.** Centroid frequency (**top**) and Shannon entropy (**bottom**) for the increasing load tests after a 300 kHz high pass filter has been applied to the signal.

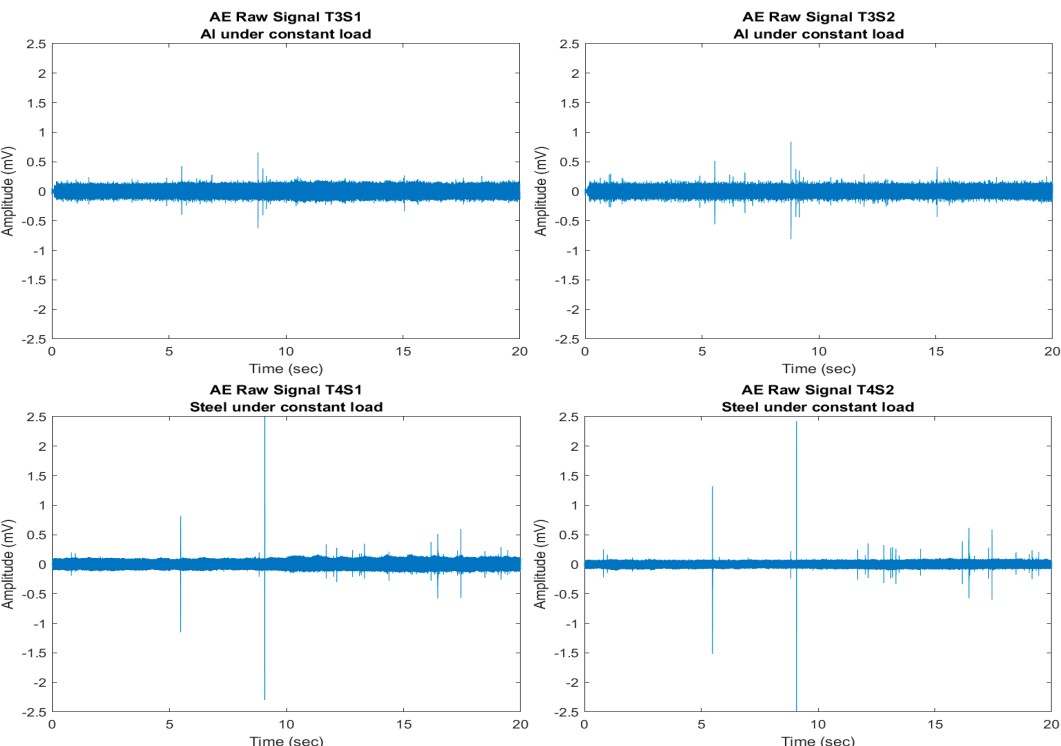

**Figure 12.** Raw AE data from the constant load tests.

The STFT spectrograms (Figure 13) for the constant load tests showed dominant frequencies in a low-frequency region (below 300 kHz), which were persistent throughout the duration of the test for all tests and are therefore attributed to machine noise. The steel/steel tribopair does not appear to show any significant activity above the noise (greater than 300 kHz) except for a few peaks at 17 s, 24 s, and 27–30 s, which correlates well with the AE events seen in the raw data, Figure 12. There appears to be much more activity in the steel/aluminium tribopair, which has been attributed to greater interaction between the indenter and workpiece/plate materials, as evidenced by the deeper penetration depth into the aluminium sheet and higher measured friction as shown in Figure 14.

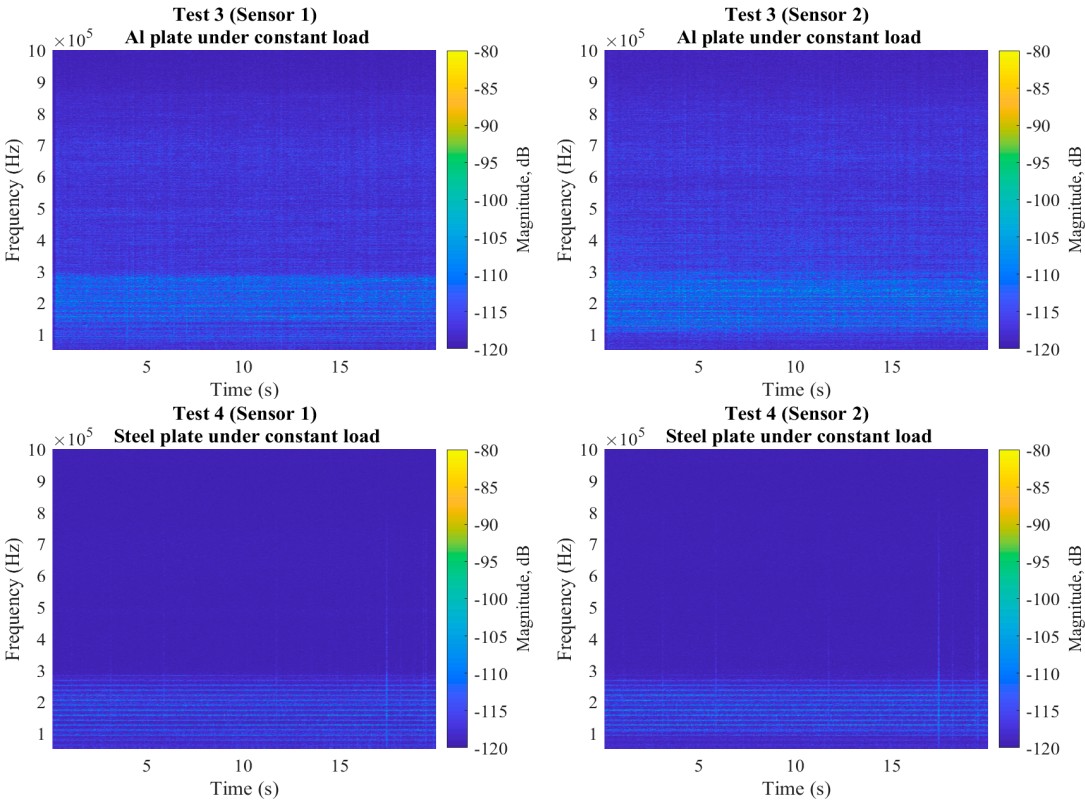

**Figure 13.** STFTs of tool steel indenter on tool steel sheet (**left**) and tool steel on Al sheet (**right**) sheet under constant load.

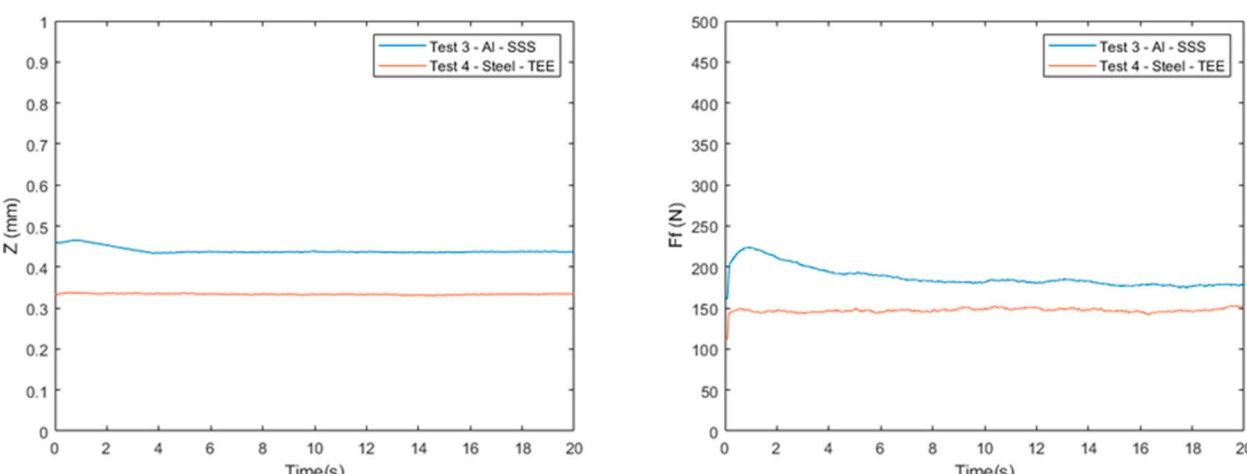

**Figure 14.** Mechanical data for tool steel indenter on tool steel sheet and tool steel on aluminium sheet under constant load. Depth of cut (**left**) and friction force (**right**).

The mechanical data show that the depth of cut was slightly greater for aluminium than steel, but remained constant throughout. The frictional force for aluminium increased sharply at the start of the test and then decreased across the duration whilst the steel remained constant, Figure 14.

The FFTs (Figure 15), again show substantial activity below 300 kHz due to the machine noise, as well as some dominant peaks at approximately 400 kHz, 500 kHz, 700 kHz, and 950 kHz for both plate materials.

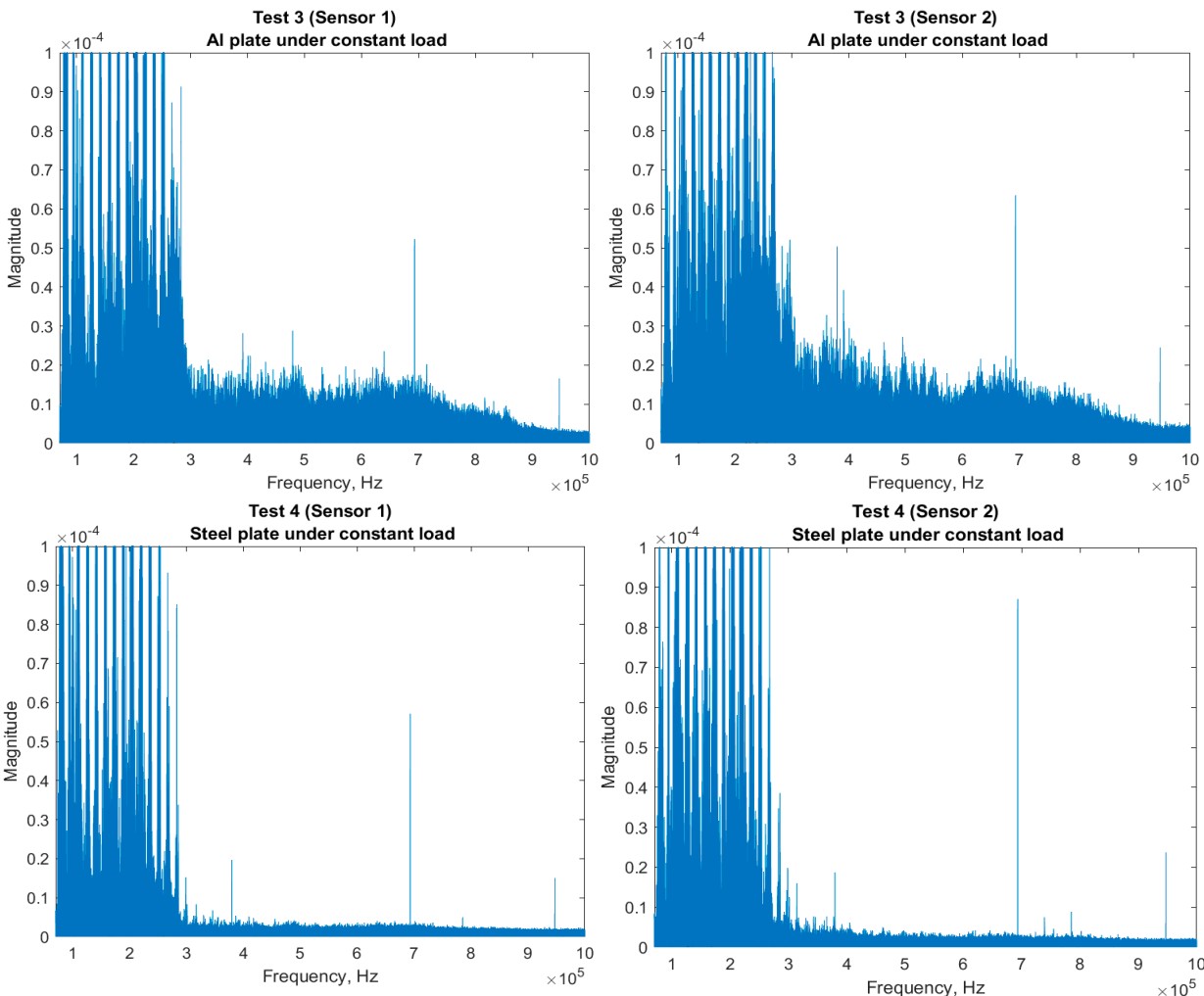

**Figure 15.** FFTs of tool steel on Al sheet (**top**) and tool steel indenter on steel sheet (**bottom**) under constant load. Left, sensor 1. Right, sensor 2.

When mean frequency is considered, Figure 16 (left) shows the steel plate samples exhibit a lower mean frequency than the aluminium plates across the duration of the test. It is difficult to see if there is a correlation between the constant load and mean frequency, as there are occasional peaks and troughs in the data. The overlap between two tests, Test 3 and Test 4, makes it difficult to comment on a difference in materials. However, once the mean frequency is recalculated after a 300 kHz high pass filter is applied to the data Figure 16 (right), the steel plate once again exhibits a higher mean frequency (600 kHz) than the aluminium (500 kHz) across the duration of the test; this is similar to what was observed in the increasing load tests.

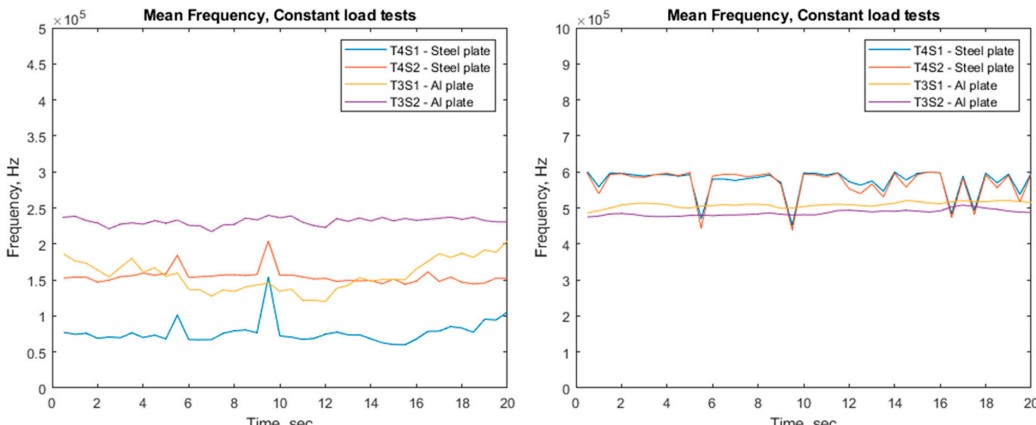

**Figure 16.** (**Left**) Mean frequency of tool steel indenter on steel sheet and tool steel on aluminium sheet under constant load including the sub 300 kHz region. (**Right**) Mean frequency of tool steel indenter on tool steel sheet and tool steel on aluminium sheet under constant load with 300 kHz high pass filter.

The centroid frequency (Figure 17, top) shows the steel typically has a higher centroid frequency (600 kHz) than the aluminium (500 kHz) throughout the duration of the constant load test, and the centroid frequencies for both materials remain almost constant. However, Shannon entropy (Figure 17 bottom) shows an increase with load for the aluminium sample not seen in the steel sample.

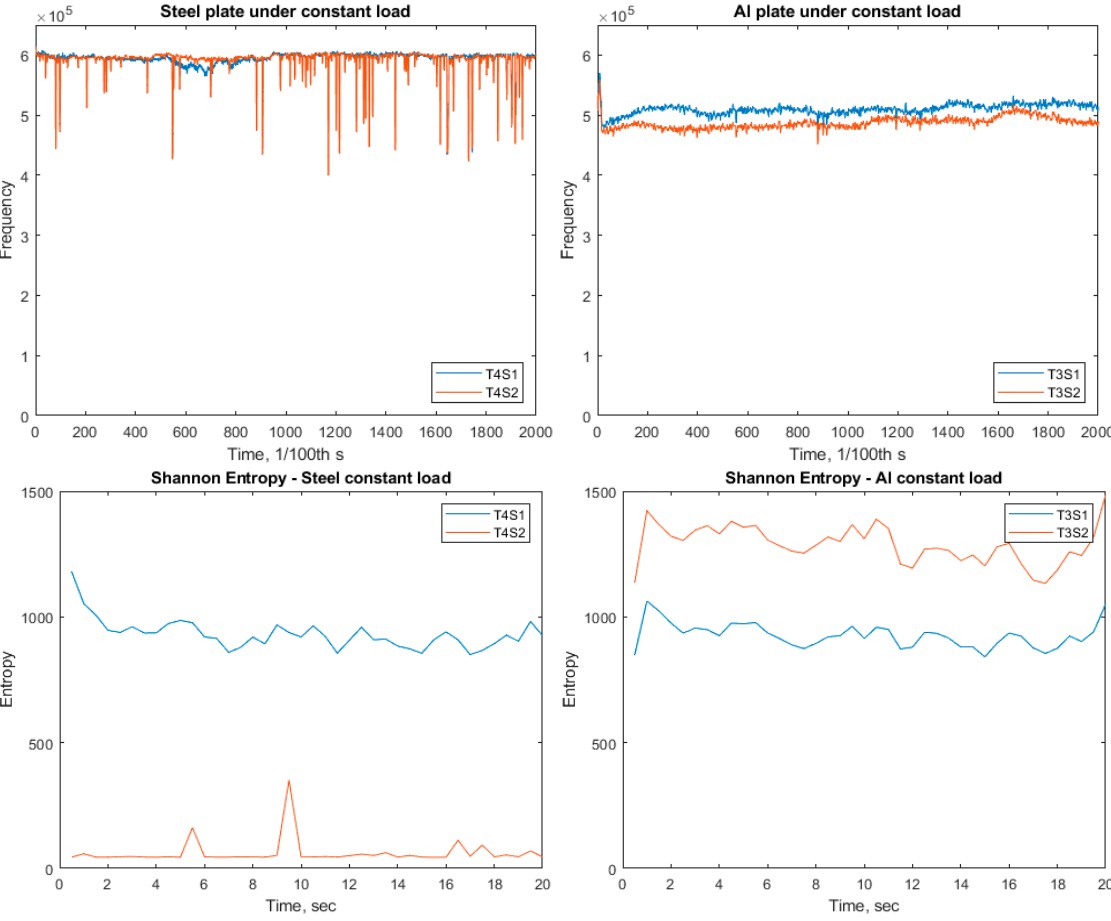

**Figure 17.** Centroid frequency (**top**) and Shannon entropy (**bottom**) for the constant load tests, after a 300 kHz high pass filter has been applied.

As described with the increasing load tests in Section 3.1, the AE hits and counts were measured, but the threshold for the hits was set too low such that noise was sufficient to trigger a hit. Therefore, these data are not considered in this paper.

Visual inspection showed that galling did not take place, evidenced by the lack of transferred material from the sample to the indenter, Figure 18. For completeness and to show that the samples were indeed scratched, an example image has been included, Figure 19. Differences are attributed to differing frictional forces between the counter face and the differing material characteristics of each sample, as discussed later.

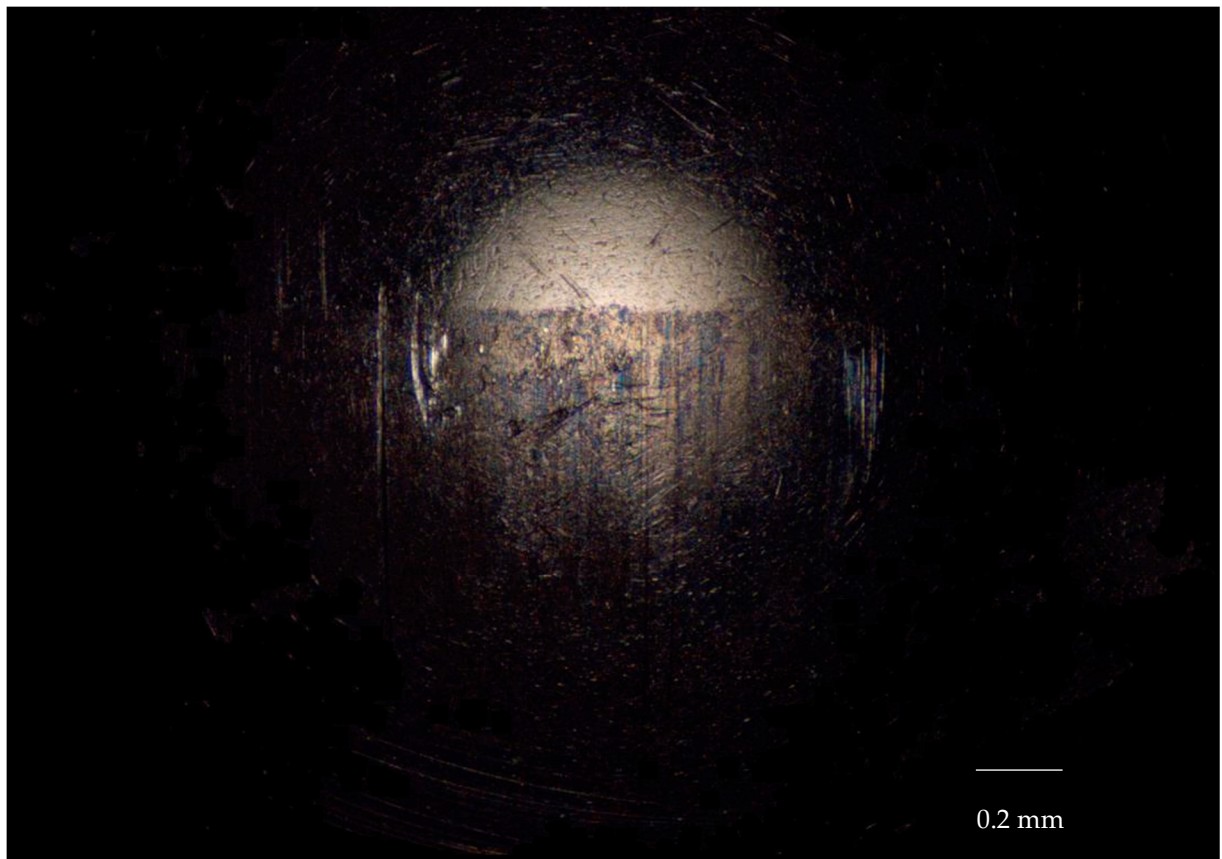

**Figure 18.** Indenter for test 4, showing no transferred material from the steel sample to the indenter.

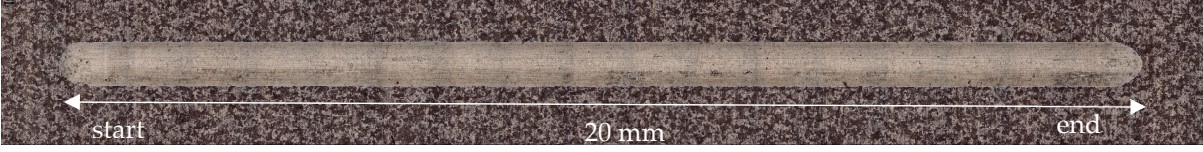

**Figure 19.** Example scratch from test 4, a steel sample.

### 3.3. Single Grit Tests

Figure 20 shows the raw data for the AE response of the four materials used in the SG tests, which display observable differences for the time–domain when considering SG and different material interactions. This is important in respect to the argument that different material interactions give off different ranges of acoustic energy.

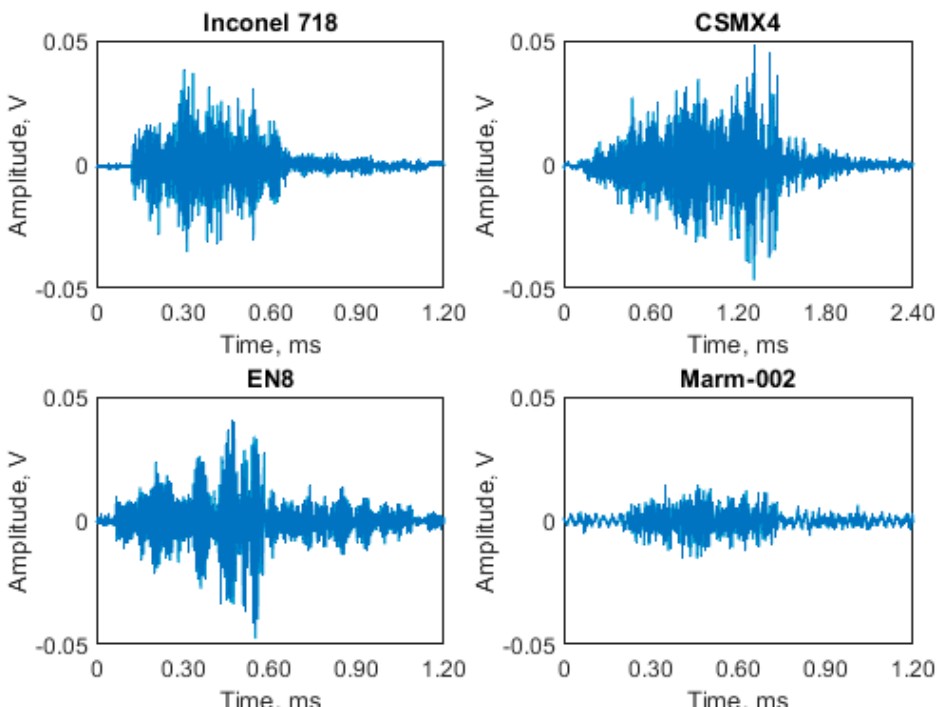

**Figure 20.** AE SG events for Inconel 718, CSMX-4, EN8 & MARM-002.

Figure 21 shows the STFTs of the AE from the SG tests. The four tests all show a clear period of activity, which must be from the scratch. As well as showing noise at 100 kHz, they each show a different frequency response, depending on the material: Inconel 718 around 200 kHz, CSMX-4 at 150 and 600 kHz, EN8 at 150 kHz, and MARM-002 at 150, 220, and 550 kHz. The AE waveform from the CSMX-4 test is notably longer and of greater amplitude than the other tests.

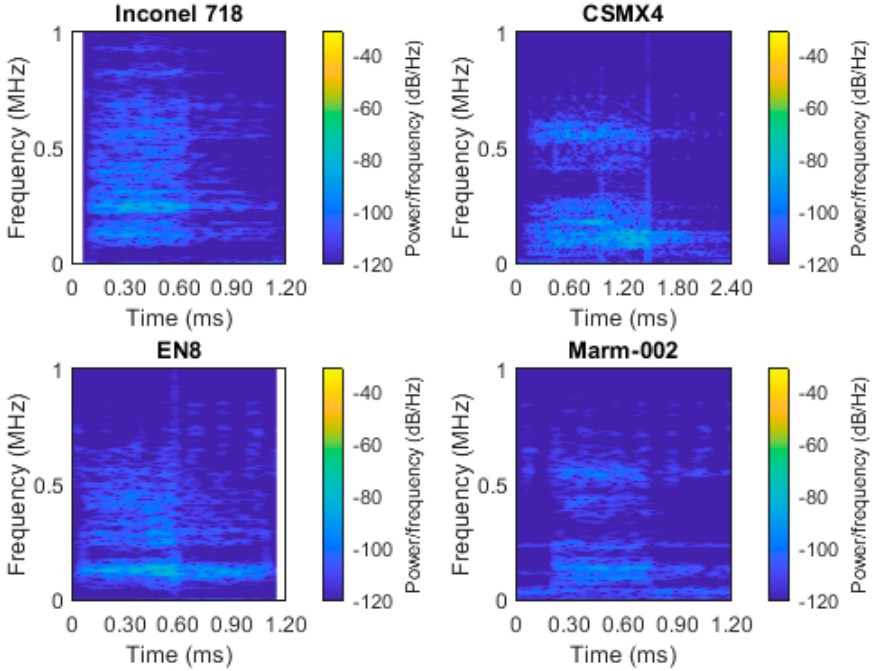

**Figure 21.** AE STFT for SG tests for Inconel 718, CSMX4, EN8, and MARM-002.

The SG tests were originally carried out to look at the micro effects of cutting, ploughing, and rubbing [21]; however, at the same time they are controllable and repeatable tests in

terms of loads and therefore provide a suitable test to display the differences of AE energy passing through different materials. The SG scratches are not a single phenomenon on the workpiece surface; instead, there are about 15 scratches in all where the fixed SG mounted to a steel disk is rotated at commercial speeds and 1μm incremented towards the workpiece. As soon as the scratches approach, the workpiece is inspected for plastic deformation. Once contact is made, smaller scratches with rubbing exist and cutting predominately exists at the middle of the workpiece. The grit signatures displayed in Figure 20 display predominately cutting phenomena as opposed to rubbing and ploughing phenomena.

This is important in respect to the argument that different material interactions give different ranges of acoustic energy and especially what these physical actions mean from an acoustic energy perspective when applied to different materials. Based on very slight inconsistencies with setup in terms of a totally flat surface, there may be more energy exerted with one test to another—i.e., where the amplitude increases as the interaction increases between the indenter and workpiece. However, the frequency bands do change dependently upon the material under test, and this is key for discussions and the underlying argument.

Figure 22 shows that each Centroid Frequency for the various SG tests is significantly different from the other. There is no notable trend for each of the materials. Figure 23 shows the Shannon entropy for each of the tests. There is an increase in Shannon entropy with time for the EN8 and CSMX4, which rapidly drops to 0 as the AE event ends, whereas the Inconel 718 shows a gradual increase and decline across the duration of the AE event. The magnitude of the MARM-002 is notably smaller than the other three materials.

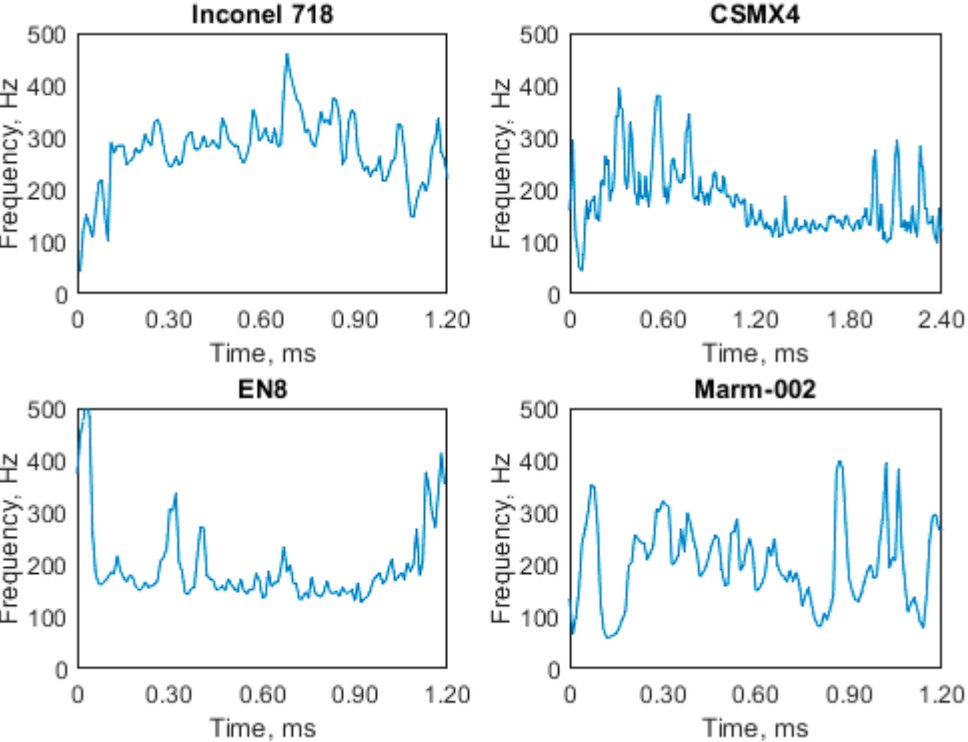

**Figure 22.** Centroid Frequency of the SG tests for Inconel 718, CSMX4, EN8, and MARM-002.

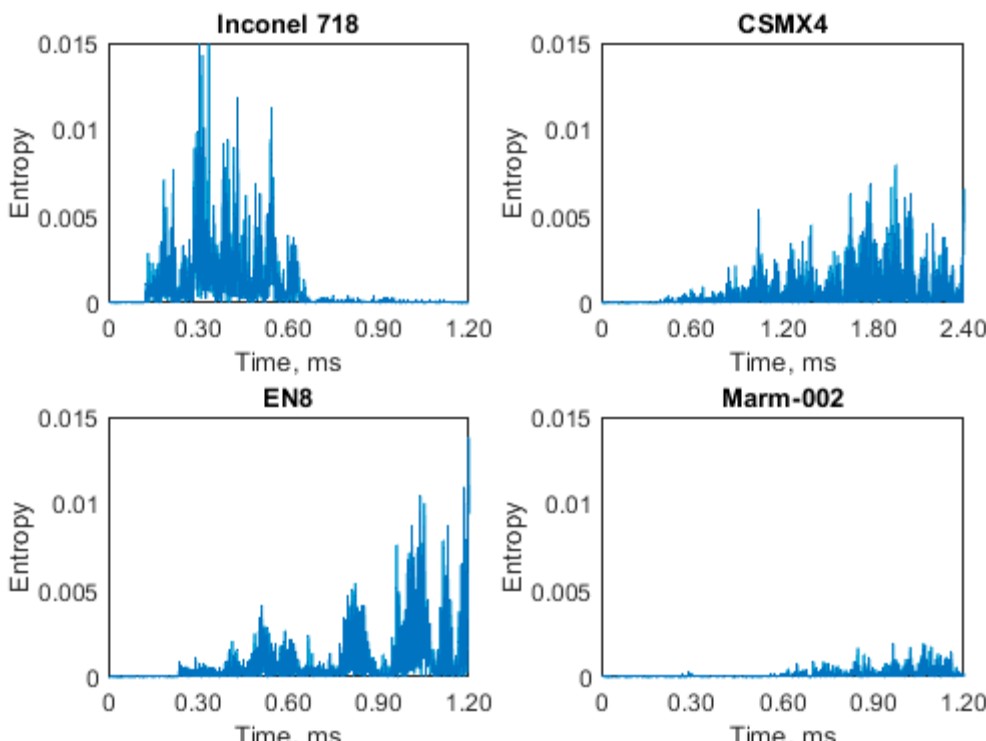

**Figure 23.** Shannon entropy of the SG tests for Inconel 718, CSMX4, EN8, and MARM-002.

## 4. Discussion

All scratch tests under both loading conditions, irrespective of plate material (aluminium or steel alloy SA52210), showed significant activity believed to be noise below 300 KHz. The FFT analysis consistently shows activity at 400 kHz, 500 kHz, 700 kHz, and 950 kHz across all tests (Figures 9 and 15).

In the STFT spectrograms (Figures 7 and 13) for both increasing and constant load, there appears to be a difference between the steel/steel tribopair and the steel/aluminium tribopair, with the latter (steel/aluminium) showing much more activity in the frequency range above 300 kHz. In comparison to the SG tests, the scratch tests exhibit much more noise.

Sindi et al. reported that stage 1 plastic deformation wear should occur in the region of 60 to 125 kHz, and Moghadam found wear at 20 kHz to 160 kHz [18,22]. Shanbhag et al. identified that adhesive wear ranged from 100 to 500 kHz and abrasive wear from 100 kHz to 200 kHz [16,19]. These works caused us to expect results below 300 kHz; however, the STFTs from our work appear to show that sub-300 kHz is dominated by noise. When looking above 300 kHz, the same distinct frequencies are shown by both materials, making them indistinguishable in the frequency domain when using Fourier analysis.

Mean frequency separates the materials, with steel plates having a lower mean frequency than aluminium in the increasing load situations (Figure 10 left) and perhaps lower in the constant load situation (Figure 16 left). However, these calculations include the activity at sub-300 kHz, which we believe is noise. Once this region is filtered out, as per Figure 10 (right) and Figure 16 (right), the steel plates exhibit a higher mean frequency (600 kHz) than the aluminium samples (500 kHz) across the duration of the scratch, which is also seen in the centroid frequency.

Sharp changes in the mean frequency and centroid frequency are also evident in the STFT spectrograms (Figures 7 and 13). However, these events in the mean frequency and centroid frequency graphs and STFT spectrograms are not shown in the mechanical data.

Centroid frequency appears more sensitive to change than the mean frequency; however, the windowing for the centroid frequency is shorter in comparison to the mean frequency (0.1 s vs. 0.5 s for the mean frequency), which may explain this difference.

The Shannon entropy showed an increase with increasing load for the aluminium sample. This increase is not seen in the corresponding increasing load test of the steel sample. It is interesting to note the large difference between the two sensors for both samples when under constant load.

Shanbhag et al. used the mean frequency technique to show a difference between worn and unworn components, but not between materials [16,19]. Shanbhag et al. reported that mean frequency outperformed other time–frequency techniques for showing signs of early wear. The results of our work confirm this conclusion of Shanbhag et al. [16,19], in that mean frequency apparently outperforms Fourier analysis by discriminating between materials not seen in the Fourier analysis.

Aluminium, as a softer material than tool steel (aluminium had a greater depth of cut and greater friction between indenter and plate for all tests), exhibits a lower mean frequency for all tests as well as a lower centroid frequency. This is believed to be due to the differing crystal structures of the materials, as microscopy showed no galling took place. Differing frictional forces between the counter face and samples attributed to the greater ductility of the aluminium alloy in comparison to the steel alloy used. In other words, the steel alloy resisted the deformation better than the aluminium, as expected, and this is reflected in the AE signals.

The SG tests seem to show greater intensities for materials with greater levels of hardness, Figure 21. EN8 steel is the only material which does not have high-intensity peaks when compared to the Ni-based alloys. This is attributed to the higher hardness of the material requiring more energy to create dislocations and deform the material. The difference between CSMX-4 and the other Ni-based materials in these tests is attributed to a deeper (and therefore longer) cut in the material, resulting in greater plastic deformation. These results are in agreement with Hase et al., Chen et al., and Griffin, whose works showed that the AE response of different materials should be different and distinguishable in the time–frequency region, and also in line with the work of Skare et al., which states a drawback of AE is that several early phenomena (cracking, movement of dislocations, twinning, movement of grain boundary, break of cohesion bindings between different layers and inclusions in the material, flow of medium, cavitation, internal and boundary friction, impacts, growth of magnetic domains, condensation, and solidifying process of material or structures) are indistinguishable [13,17,20,21].

For the SG tests, the Centroid frequency and Shannon entropy techniques again appear to discriminate between some materials. For Shannon entropy, a difference in EN8 and CSMX4 is hard to spot; however, the Inconel 718 and MARM 002 are different in shape and magnitude. The Centroid frequency analysis reflects this, with the EN8 and CSMX4 appearing very similar to one another.

## 5. Conclusions

This research set out to investigate the AE response of different materials when there is insufficient load to initiate galling and compare the scratch tests to single grit tests. This research found the following for scratch tests:

- Significant frequencies for the tests were found at far higher values than expected, at 400 kHz, 500 kHz, 700 kHz, 800 kHz, and 950 kHz. The expected frequency range, 60 kHz to 125 kHz, was found to be dominated by noise.
- There was no difference found between the AE signature of the aluminium or the steel for indenter scratch tests in the frequency domain.
- The mean frequency, centroid frequency, and Shannon entropy parameters showed a difference not reflected in Fourier techniques. Steel had a higher mean and centroid frequency (600 kHz) than aluminium (500 kHz), attributed to a difference in hardness, as demonstrated by a difference in depth of cut.
- Mean frequency performs as well as centroid frequency (for scratch tests).
- Visual inspection showed that galling did not occur, so differences are mostly attributed to differing frictional forces between the counter face and samples.

- In addition, this research found the following for single grit scratch tests:
- From the SG tests, the NI-based alloys appear to exhibit a distinguishable AE signature from one another in terms of frequency response.
- The frequency responses for the SG tests were found in the expected region, unlike the indenter scratch tests.
- Looking further into the SG tests, for each individual material there are differences in terms of AE response, This needs to be taken into consideration when carrying out scratch tests and using AE to correlate damage mechanisms.
- Centroid frequency and Shannon entropy can be used to discriminate between some materials.

**Author Contributions:** Formal analysis, T.D.; Investigation, B.R. and M.P.; Methodology, J.M.G.; Software, T.D.; Supervision, B.R., M.P. and J.M.G.; Validation, T.D.; Writing—original draft, J.M.G.; Writing—review & editing, T.D., B.R. and M.P. All authors have read and agreed to the published version of the manuscript.

**Funding:** This research received no external funding.

**Informed Consent Statement:** Not applicable.

**Data Availability Statement:** Data will be available upon request to the corresponding author.

**Conflicts of Interest:** The authors declare no conflict of interest.

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
