# Peer review of "Analysis of Acoustic Emissions for Determination of the Mechanical Effects of Scratch Tests"

_applsci, doi:10.3390/app12136724_

Round 1

Reviewer 1 Report

The present paper proposes an Analysis of acoustic emission emissions during scratch tests to study damage during sheet metal forming. The authors performed two set of experiment (scratch tests and single grid tests) on different couple of materials and analysed the acoustic emission mostly in term of frequency of the events. The choice of such experimental is not justified at all in the manuscript and it is far to be obvious that scratch test is relevant for studying metal forming.

Unfortunately, the experimental set-up and conditions are very poorly described and sometime confusing. As an example from many others, figure 3 caption is far to be instructive enough, the different denomination for the sensors re not explain, the number of sensors used is not clear (1,4 6 or even more?). on line 114, there is “Error! Reference source not found” which is not acceptable when a manuscript is submitted to an editor. On L 112, “positions of the sensors were moved between tests to also measure this effect, is really not clear for the reader…

The description of the results is again far to be of sufficient level to be published. A lot of information are missing to understand the results. Again as some examples extracted from too many others, the duration of the scratch test with increasing load is not mentioned and thus it is impossible to link the AE analysis with the loading. Figure 7 is very difficult to read. On figure 6 and 12, it is very difficult to see any peak or activity except the noise band at low frequency. By the way, there is no explanation from the origin of this noise. On figure 12, one can obviously see large vertical band which are not described in the manuscript. It is very strange to see constant evolution of the AE parameters both in constant and increasing load scratch test as if the load as no effect on the results. The single grid test re very poorly described and the results are even more poorly analysed. What is the use of figure 18? Why do they perform such tests on different materials? Finally, for both scratch and grid test, the link between the AE and the observed damaged is missing. Having this link is, in my opinion, which the only goal of such paper.

Finally, for all the above reasons and for many others, I don’t think that this paper is ready for publication.

Author Response

To Reviewer,

Please accept our sincere apologies for the delay as it's been quite a challenge to communicate from the extremes of the planet (Australia - UK). We would like to thank you for your understanding and kind actions in giving us extra time. 

I will be uploading both the track changed manuscript document and the reviewers' document that addresses comments to all three reviewers. Please make sure that all three reviewers get a copy of both files to make a fair and easy assessment.

Kind regards

James Griffin

Detailed response to reviewers’ comments and revisions

Manuscript ID: applsci-1739405
Title: Analysis of acoustic emissions for determination of the mechanical effects made during scratch tests
Authors: Timothy Michael Devenport, Bernard F. Rolfe, Michael Pereira, James Marcus Griffin *

We, the authors of the abovementioned paper, would like to thank the editor and reviewers for the review of our paper.  The constructive and insightful comments from all three reviewers have enabled us to improve the quality of our manuscript.

This document details our point-by-point response to each of the comments provided by the reviewers.  We have also described the changes that we have made to the manuscript, which has been revised using the “Track Changes” tool in Microsoft Word, as requested.  The line numbers that we refer to in the last two columns of the table below, refer to the line numbers in the Word document when “All Markup” is shown.

Reviewer Number 1:

Comment number

Reviewers’ comment

Authors’ response

Changes in the manuscript

1.1

The present paper proposes an Analysis of acoustic emission emissions during scratch tests to study damage during sheet metal forming.

The authors performed two set of experiment (scratch tests and single grid tests) on different couple of materials and analysed the acoustic emission mostly in term of frequency of the events.

The choice of such experimental is not justified at all in the manuscript and it is far to be obvious that scratch test is relevant for studying metal forming.

Scratch tests have been used in other work in the literature to study wear mechanisms in metal forming.  Additionally, the single grit tests are used in this work as a secondary experimental setup to display the differences of acoustic emission (AE) when grit and different materials interact. The results from the comparison of these two tests and different materials, allows for further insights into the acoustic emissions response.

Based on the reviewer’s comments, we have added additional explanation in the Experimental Methods section regarding this.

At line 102, we have added the following:

Scratch testing has been used to investigate the AE response during galling wear for metal forming applications in other works in the literature (Heinrichs, Olsson, and Jacobson 2012; Shanbhag et al. 2018; Shanbhag et al. 2018; van der Heide and Schipper 2003).

At line 105, we have added the following:

SG testing has been used to display the differences of AE when grit and different materials interact. This is important as AE ranges can differ between different material phenomena interactions.

1.2

Unfortunately, the experimental set-up and conditions are very poorly described and sometime confusing.

Upon further review, we agree with the reviewer’s comments and so we have extensively revised the Experimental Methods section to improve the clarity and provide further details and description.

We have made major revisions to Section 2 (from line 102 to line 203).  Please refer to the Tracked Changes in this section of the revised manuscript, as there are too many changes to list here.  

1.3

As an example from many others, figure 3 caption is far to be instructive enough, the different denomination for the sensors re not explain, the number of sensors used is not clear (1,4 6 or even more?).

As per our response to Comment 1.2, we have significantly revised the Experimental Methods section.  The sensor setup is more clearly explained (see lines 156 to 161). The Figure 3 caption (now Figure 5) is updated to be more descriptive/instructive.

As described in item 1.2 above, we have made major revisions to Section 2 (from Line 102 to Line 203). 

The sensor setup is described in the text added at Line 154:

The positions of the sensors were moved between tests to observe the effect the positions of the sensors have on the measured AE signal. These positions were on the scratch face parallel to the scratch direction, on the scratch face perpendicular to the scratch direction, on the “edge” of the sample parallel to the scratch direction, and finally on the “edge” of the sample perpendicular to the scratch direction. A schematic of the sensor positions is given in Figure 5. It must be stressed that only 2 AE sensors were used during any one test, as indicated by the sensor configuration described in Table 1.

At line 161, the update Figure 3 caption (now Figure 5) is:

Figure 5 – Schematic of test sensor positions. A = on the scratch face parallel to the scratch direction (red markers). B = on the “edge” of the sample perpendicular to the scratch direction (green markers). C = on the “edge” of the sample parallel to the scratch direction (yellow markers). D = on the scratch face perpendicular to the scratch direction (orange markers). Note measurements from orientation D are not used in this work.  

1.4

on line 114, there is “Error! Reference source not found” which is not acceptable when a manuscript is submitted to an editor.

We have corrected this.

Corrected.

1.5

On L112 “positions of the sensors were moved between tests to also measure this effect, is not really clear for the reader…

We have reworded this to improve the clarity for the reader.  Please refer to our response and text added in item 1.3.

This reworded text is now at line 156.  Refer to item 1.3 above, for the reworded and added text.

1.6

The description of results is again far to be of sufficient level to be published. A lot of information are missing to understand results.

We have added more information to describe the figures with a literal observation of what the figures show and what we want the reader to see in the figures.

This information is then used in discussion.

All figures provided with accompanying literal observation of figures and discussed in the discussion. Again, there are too many changes to list here.  Please refer to the Tracked Changes in Section 3 (Results) of the revised manuscript (line 208 to line 397).

1.7

Again as some samples extracted from too many others, the duration of scratch tests with increasing load not mentioned and thus it is impossible to link the AE analysis with the loading.

The duration of the scratch tests is now given in table 1, and the figures are all correctly scaled to match this.

The relationship between the applied load and the AE data is more clearly linked in the discussion.   

Duration of scratch has been added as an additional column in Table 1 (line 120). The graphs have been rescaled to only show duration of scratch – see for example Figures 7, 9, 13, 15 in the revised manuscript.

As per item 1.6, we have made significant changes to the explanation of the results in Section 3, where the linking between the AE signal and the loading is more clearly explained.  Additionally, the loading condition is included as a title in all of the figures for added clarity – as shown in figures 6, 7, 8, 9, 10, 12, 13, 14, 15, 16 .

1.8

Fig 7 difficult to read.

We have reproduced some figures to make them more read-able for the reader.

Figure 7 (now Figure 8) is changed to multiple separate FFT graphs to prevent overlap of data.  The same change was applied to Figure 14 to improve the clarity.

1.9

On Figure 6 and 12 it is very difficult to see any peak or activity except the noise band at low frequency.

We believe this was due to a graphics issue and have reproduced the STFTs to produce higher quality image.

The STFT’s images have been replaced, see figures 7 and 13.

1.10

By the way, there is no explanation for noise.

We have added an explanation for the noise, believed to be machine noise as constant in all tests.

The following text has been amended to explain the noise (line 217 and line 290):

“… the STFT spectrogram (Figure 7) shows dominant frequencies in a low frequency region below 300 kHz were persistent throughout the duration of all the tests and are therefore believed to be machine noise as this is seen in all tests.”

The STFT spectrograms (Figure 13) for the constant load tests showed dominant frequencies in a low frequency region (below 300 kHz) which were persistent throughout the duration of the test for all tests and are therefore attributed to machine noise.

1.11

On Fig 12, one can obviously see large vertical band which are not described in the manuscript.

We believe this vertical band was due to either the graphics error (as per comment 1.9) or an artefact from outside the scratch duration (see comment 1.7) as it is no longer present in the reproduced STFTs.

Large vertical bands are no longer present in the revised STFT figures – see Figure 13.

1.12

It is very strange to see constant evolution of AE parameters both in constant and increasing load scratch tests as if the load has no effect on results.

The loads are insufficient to generate galling wear therefore there is very little change in the STFTs for these types of scratch tests. This is particularly evident for the steel tests, where the noise is dominant.  For the aluminium tests, an increase in trend in the signal is evident.  We have provided the additional raw AE data graphs (see Figures 6) and additional AE parameters (centroid frequency and Shannon entropy, Figure 10).  These illustrate the affect of the increasing load on the AE results. 

In terms of comparing the two materials, the observable differences are significant and this is the underlying message within this work, i.e. that the setups using different materials experience different frequency ranges of acoustic emission. The acoustic emission amplitude increases if the workpiece and indenter increase in terms of interaction. 

Additional figures (Figures 6, 10) show the effect of the increasing load on the AE results. 

The following text was added to explain these results (line 239):

“… the centroid frequency for the steel remains almost constant whilst the aluminium exhibits a very gentle increase in centroid frequency. However, Shannon entropy (Figure 10 bottom) shows an increase with load for the Aluminium sample not seen in the steel sample.”

1.13

The single grit test re very poorly described and the results are even more poorly analysed.

Upon further review, we agree with the reviewer’s comments and so we have revised the single grit test description and provided further explanation and analysis of the single grit test results.

We have made major revisions to Section 3.3 (line 370 to 397) to describe the single grit test results.  Please refer to the Tracked Changes in this section of the revised manuscript, as there are too many changes to list here.   

1.14

What is the use of figure 18?

Figure 18 (now Figure 20) shows the shows the raw data for the AE response of the four materials used the single grit tests.  This, in conjunction with the STFTs in Figure 21, display observable differences for the time-domain when considering SG and different material interactions.  Based on the reviewer’s question, we have provided additional explanation in the text to explain why this is relevant.  

The following text is added/amended to explain Figure 18 (line 371):

“Figure 20 shows the raw data for the AE response of the four materials used the single grit tests which displays observable differences for the time-domain when considering SG and different material interactions. This is important in respect to the argument that different material interactions give off different ranges of acoustic energy.”

1.15

Why do they perform tests on different materials?

The tests are performed on different materials to show the differences of AE when grit and difference materials interact.

The following text has be added to clarify the reasons why we performed tests on different materials (line 107):

“SG testing has been used to display the differences of AE when grit and different materials interact. This is important as AE ranges can differ between different material phenomena interactions.”

Also (line 392):

“This is important in respect to the argument that different material interactions give different ranges of acoustic energy and especially what these physical actions mean from an acoustic energy perspective when applied to different materials. Based on very slight inconsistencies with setup in terms of a totally flat surface, there may be more energy exerted with one test to another. The amplitude increases as the interaction increases between the indenter and workpiece.  However, the frequency bands do change dependent on the material under test, and this is key for discussions and the underlying argument.”

1.16

Finally, for both scratch and grit tests link between AE and observed damage is missing.

(Having link is, in my opinion, which the only goal of this paper.)

As described, this paper is focused on the conditions for “pre-galling” operating conditions. The scratch tests were conducted at loads too low to generate galling, because the focus is on the influence of the particular set-up and the materials and geometries used on the signal prior to galling wear on the indenter. The typical geometry associated with the scratch on the sample and indenter is shown in Figures 18 and 19.   

The work cited from Griffin (2015) for the single grit tests displays a scratch profile which is in regard to the elastic and plastic deformation of the material surface. Depending on the phenomenon either more built-up edge would be apparent for ploughing when compared with scratch depth for cutting. Rubbing is observed if the duration of the acoustic emission signal is greater in length than the profile physical measurement.   

No changes, as this was not the focus of this work.  Interested readers can refer to the work from Griffin (2015) for differences in the scratch geometry during the different stages of grit interaction.

Griffin, J. (2015) 'Traceability of Acoustic Emission Measurements for a Proposed Calibration Method – Classification of Characteristics and Identification using Signal Analysis'. Mechanical Systems and Signal Processing 50-51, 757-783

Reviewer 2 Report

This paper investigated acoustic emission signals of metallic materials during scratch tests. The topic of this paper is interesting. However, the results and discussion sections are relatively weak. Before publication, following shortages must be overcomed.

  1. The title (Analysis of acoustic emission emissions for….) should be revised.
  2. Some spelling errors such as Line 94 “however it isexpected ” need careful check.
  3. The authors stated that from literature the frequency region of AE signals generated during scratching is expected to be sub 300 kHz. However, the major frequency region of AE obtained in this study is much higher. More analyses should be performed in the Discussion section to compare your results with literature.
  4. In Line 260, the authors stated “They are all of similar magnitude and duration, although the shape differs between materials.” However, from Figure 18, some differences among the four typical waveforms are evident. In particular, the peak amplitude of the AE signal during SG test for Marm-02 is much lower than others, and the duration of waveform of SG test for CSMX4 is much longer than others (longer than 3000 ms). Why the authors neglected these differences?
  5. This paper extracted mean frequency and count from AE signals to characterize the damage process of different materials. However, as a research paper, the only use of two parameters is not enough. Actually, a number of commonly used parameters can be extracted from AE waveforms such as time domain parameters including amplitude, entropy, count, duration, rise time, energy, etc, and frequency domain parameters including the peak frequency, frequency centroid, etc. The use of multiple parameters is recommended for eliminating the error in using only one parameter. The following papers regarding the damage characterization by multiple parameters are recommended to be cited. Moreover, the authors need to illustrate the advantages or reasons of using mean frequency, count, and STFT method. Also, if possible, I recommend the authors calculate more parameters to provide further supports for your results.

https://doi.org/10.1016/j.engfracmech.2020.107083

https://doi.org/10.1016/j.ijfatigue.2022.106860

http://dx.doi.org/10.1016/j.ymssp.2017.08.007

Author Response

Detailed response to reviewers’ comments and revisions

Manuscript ID: applsci-1739405
Title: Analysis of acoustic emissions for determination of the mechanical effects made during scratch tests
Authors: Timothy Michael Devenport, Bernard F. Rolfe, Michael Pereira, James Marcus Griffin *

We, the authors of the abovementioned paper, would like to thank the editor and reviewers for the review of our paper.  The constructive and insightful comments from all three reviewers have enabled us to improve the quality of our manuscript.

This document details our point-by-point response to each of the comments provided by the reviewers.  We have also described the changes that we have made to the manuscript, which has been revised using the “Track Changes” tool in Microsoft Word, as requested.  The line numbers that we refer to in the last two columns of the table below, refer to the line numbers in the Word document when “All Markup” is shown.

Reviewer number 2:

Comment number

Reviewers’ comment

Authors’ response

Changes in the manuscript

2.1

Title should be revised

We acknowledge the typo of the repeated word “emission” in the title, and so we have revised this.

Updated title (line 2):

“Analysis of acoustic emissions for determination of the mechanical effects made during scratch tests”

2.2

Some spelling errors such as line 94 “however it isexpected” need careful check

We acknowledge there were multiple errors in the document and have carefully checked the revised document.

The revised manuscript contains many corrections to address this.

2.3

The authors stated that from literature the frequency region of AE signals generated during scratching is expected to be sub 300 kHz. However, the major frequency region of AE obtained in this study is much higher. More analysis should be performed in the discussion section to compare your results with the literature.

Whilst we explain in the introduction we expected the frequency region of interest to be sub 300 kHz, we found the activity below 300 kHz in this work to be machine noise as evidenced by the Fourier analysis, which showed this activity was consistent across all tests.

We have amended the document to show how this observation was made, by including the sub 300 kHz signals in the FFT and STFT figures, as well as shown how filtering this data affects the Mean frequency and how this aligns with the Shannon Entropy and Centroid frequency.

FFT, STFT and Mean frequency graphs changed to include the sub 300 kHz region, figures 7,8 & 9 as well as figures 13, 14 &15.

As per item 1.10 above, the following text has been amended to explain the noise (line 217 and line 290):

“… the STFT spectrogram (Figure 7) shows dominant frequencies in a low frequency region below 300 kHz were persistent throughout the duration of all the tests and are therefore believed to be machine noise as this is seen in all tests.”

The STFT spectrograms (Figure 13) for the constant load tests showed dominant frequencies in a low frequency region (below 300 kHz) which were persistent throughout the duration of the test for all tests and are therefore attributed to machine noise.

2.4

L260, authors stated “they are all of similar magnitude and duration, although the shape differs between materials” However figure 18 some differences evident. In particular, peak amplitude of Marm 002 is much lower than others, and the duration of csmx4 is much longer than others (longer than 3000 ms). Why did authors neglected these differences?

We have provided additional text to explain the reasons for this difference.  Looking at these STFTs it is more about the differences of AE frequency for a given material and this is one of the main key findings within this work. The amplitude increases as the interaction increases between the indenter and workpiece. 

The following additional text has been added at line 379:

“The AE waveform from the CSMX-4 test is notably longer and of greater amplitude than the other tests.”

And line 394:

“Based on very slight inconsistencies with setup in terms of a totally flat surface, there may be more energy exerted with one test to another. The amplitude increases as the interaction increases between the indenter and workpiece.  However, the frequency bands do change dependent on the material under test, and this is key for discussions and the underlying argument.”

2.5

This paper extracted mean frequency and count from the AE signals to characterize the damage process of different materials.

However, as a research paper, only 2 parameters is not enough.

Actually a number of commonly used parameters can be extracted from AE waveforms such as time domain parameters including amplitude, entropy, count, duration, rise time, energy, etc.

The use of multiple parameters is recommended for eliminating error in using only one parameter.

We find the reviewers comment very helpful and informative. Typically, this many parameters are not used when measuring galling (as per the papers that we have referenced: Heinrichs, Olsson, and Jacobson 2012; Shanbhag et al. 2018; Shanbhag et al. 2018; van der Heide and Schipper 2003). However, as per the papers suggested by the reviewer in comment 2.6, AE is much more extensively researched in applications to composite failure and structural health monitoring than in wear detection.

As a result of this we have included Shannon Entropy and Centroid frequency in our analysis.

We have referencing other papers where this many parameters are used for galling wear analysis (line 103):

Scratch testing has been used to investigate the AE response during galling wear for metal forming applications in other works in the literature (Heinrichs, Olsson, and Jacobson 2012; Shanbhag et al. 2018; Shanbhag et al. 2018; van der Heide and Schipper 2003)”

Based on the reviewer’s comments, we have incorporated Shannon entropy and Centroid frequency for comparison against mean frequency.  Please refer to the new results that we have added (Figures 10 and 12) and the associated explanation and analysis of these.  Refer also to comment 2.7 for details of the additional changes in the manuscript.

Note that due to the changes in the analysis, counts were removed as explained in the document (line 243 and line 322).

2.6

The following papers regarding the damage characterization by multiple parameters are recommended to be cited.

Moreover, the authors need to illustrate the advantages or reasons of using mean frequency, count and STFT method.

https://doi.org.10.1016/j.engfracmech.2020.107083

https://doi.org.10.1016/j.ijfatigue.2022.106860

https://dx.org.10.1016/j.ymssp.2017.08.007

We have updated the methods section to elaborate on both the types of AE parameters and choice of AE parameters in this work.

As per our response to comment 2.5 above, we have also added the analysis of additional parameters.

Added the explanation at line 168:

 “It is known from the literature that it is critically important to select the most appropriate AE parameters (Chai et al. 2022) to minimise the probability for error. It is important to select some parameters that are a function of the peak voltage (which may be influenced by the researchers’ choice of AE set-up) and some that are waveform dependent and therefore independent of the AE set up (Barile et al. 2020).

Whilst the study of the application of AE is quite advanced in the study of failure modes in fibre reinforced plastics and structural health monitoring (Chai et al. 2022), where parameters such as frequency, amplitude, duration, rise time, peak amplitude, energy, counts, centroid frequency, weighted peak frequency, partial power, number of hits & counts per events as described by Barile (Barile et al. 2020) are commonly used. Only a select few of these parameters have been used in the application of AE to galling wear. (Hase, Wada, and Mishina 2008; Hase, Mishina, and Wada 2012; Hase, Mishina, and Wada 2013; Shanbhag et al. 2018; Shanbhag et al. 2018; Sindi, Najafabadi, and Salehi 2013).

2.7

Also if possible, I recommend the authors calculate more parameters to provide further supports for your results.  

As per comment 2.5, we have included more parameters in the analysis.

As per item 2.5, we have incorporated Shannon entropy and Centroid frequency for comparison against mean frequency.  Please refer to the new results that we have added (Figures 10 and 12) and the associated explanation and analysis of these.

 Line 238:

“The centroid frequency (Figure 10 top) shows the steel typically has a higher centroid frequency (600 kHz) than the aluminium (approx. 500 kHz) throughout the duration of the increasing load test, however the centroid frequency for the steel remains almost constant whilst the aluminium exhibits a very gentle increase in centroid frequency. However, Shannon entropy (Figure 10 bottom) shows an increase with load for the Aluminium sample not seen in the steel sample.”

 Line 317:

 “The centroid frequency (Figure 16 top) shows the steel typically has a higher centroid frequency (600 kHz) than the aluminium (500 kHz) throughout the duration of the constant load test, the centroid frequencies for both materials remain almost constant. However, Shannon entropy (Figure 16 bottom) shows an increase with load for the Aluminium sample not seen in the steel sample.”

Line 439:

 “[…] which is also seen in the centroid frequency.

Sharp changes in the mean frequency and centroid frequency are also evident in the STFT spectrograms (Figure 7 & Figure 13) however these events in the mean frequency and centroid frequency graphs and STFT spectrograms are not shown in the mechanical data.

Centroid frequency appears more sensitive to change than the Mean frequency, however the windowing for the centroid frequency is shorter in comparison to the mean frequency (0.1s vs 0.5s for the mean frequency) which may explain this difference.

The Shannon entropy showed an increase with increasing load for the aluminium sample. This increase is not seen in the corresponding increasing load test of the steel sample. It is interesting to note the large difference between the two sensors for both samples when under constant load.

Reviewer 3 Report

1,  The work is interesting. However, the novelty or contribution of current work should be elaborated. 

2,There is an error on page 4, line 114.

3,How to obtain the mean frequency, or how to define the mean frequency in fig.8?  counts ?

4,Are these AE waves or Ae events in Fig.18?

5,“the mean frequency technique to show a difference between worn and unworn components, but not between materials.” details are not be founded.

Author Response

Detailed response to reviewers’ comments and revisions

Manuscript ID: applsci-1739405
Title: Analysis of acoustic emissions for determination of the mechanical effects made during scratch tests
Authors: Timothy Michael Devenport, Bernard F. Rolfe, Michael Pereira, James Marcus Griffin *

We, the authors of the abovementioned paper, would like to thank the editor and reviewers for the review of our paper.  The constructive and insightful comments from all three reviewers have enabled us to improve the quality of our manuscript.

This document details our point-by-point response to each of the comments provided by the reviewers.  We have also described the changes that we have made to the manuscript, which has been revised using the “Track Changes” tool in Microsoft Word, as requested.  The line numbers that we refer to in the last two columns of the table below, refer to the line numbers in the Word document when “All Markup” is shown.

Reviewer number 3:

Comment number

Reviewers’ comment

Authors’ response

Changes in the manuscript

3.1

The work is interesting. However, the novelty or contribution of current work should be elaborated

We have provided this explanation in our original manuscript. Based on the reviewer’s comment, we have provided some additional text to further explain this. 

The additional sentence was added to the following last paragraph of the Introduction (Section 1) at line 99:

“None of these works report on the AE response during “normal” or “pre-galling” operating conditions, which will be key for developing the AE technology and techniques for detecting and reacting to the development of galling wear. Therefore, this research uses scratch testing to investigate the AE response of different materials when the applied load is insufficient to produce galling. The results show that the AE frequencies and behavior are specific to the particular setup and the materials/geometries used.”

3.2

There is an error on page 4 line 114

We have amended the document to fix this specific error among many others as per previous comments 1.4 and 2.2.

Corrections made, as per previous comments 1.4 and 2.2.

3.3

How to obtain the mean frequency, or define the mean frequency in fig 8? Counts?

Whilst this was given in the original version, we acknowledge it may have been easily missed. This is now at line 185 in the revised manuscript.  We have also provided some additional information. If the reviewer’s comment refers to how the built in MATLAB function works, we have added a link to the MATLAB documentation in a footnote beneath line 188.

Line 182:

“Mean frequency was calculated by windowing the signal into windows of 0.5s, and calculating the mean in each window using MATLAB’s meanfreq function2. The sampling rate for the FFT, STFT and Mean frequency was set to 2 MHz as a compromise between preventing aliasing and capturing an unmanageably large data set.”

Footnote linking to MATLAB’s help for mean frequency added below line 188.

3.4

Are these AE waves or AE events in fig 18?

These are AE events where the SG interacts between different materials (now Figure 20).  We have provided additional text to explain this.

Additional text explaining SG test results provided at line 394:

“The SG scratches are not a single phenomenon on the workpiece surface, instead there is about 15 scratches in all where the fixed mounted SG to a steel disk is rotated at commercial speeds and 1µm incremented towards the workpiece. As soon as the scratches approach, the workpiece is inspected for plastic deformation. Once contact is made, smaller scratches with rubbing with cutting predominately existing at the middle of the workpiece. The grit signatures dis-played in Figure 20 display predominately cutting phenomena as opposed to rubbing and plough phenomena.”

3.5

“The mean frequency technique to show a difference between worn and unworn components, but not between materials” details are not be founded

The full sentences to which the reviewer is referring to is now at line 453:

“Shanbhag et al. used the mean frequency technique to show a difference between worn and unworn components, but not between materials. Shanbhag et al. reported that Mean frequency outperformed other time frequency techniques for showing signs of early wear. The results of this work confirm this conclusion of Shanbhag (Shanbhag, Rolfe, Arunachalam et al. 2018b).”

We appreciate that this sentence was not very clear as our work did not compare worn to un-worn components as Shanbhag et al. had.  Shanbhag et al. showed that Mean frequency outperformed Fourier analysis in differentiating unworn and worn components. Rather, we are commenting on how Fourier analysis did not discriminate materials, however Mean frequency, along with Centroid frequency and Shannon Entropy, do discriminate these materials, and may therefore be considered superior to Fourier analysis.  

We clarified the conclusions from our mean frequency analysis clarified at line 455: 

“The results of this work confirm this conclusion of Shanbhag et al. (Shanbhag et al. 2018), in that mean frequency apparently outperforms Fourier analysis, by discriminating between materials not seen in the Fourier analysis.” 

The Mean frequency is also compared against centroid frequency and Shannon entropy as per comment 2.7, at line 441:  

 “[…] which is also seen in the centroid frequency.  

Sharp changes in the mean frequency and centroid frequency are also evident in the STFT spectrograms (Figure 7 & Figure 13) however these events in the mean frequency and centroid frequency graphs and STFT spectrograms are not shown in the mechanical data. 

Centroid frequency appears more sensitive to change than the Mean frequency, however the windowing for the centroid frequency is shorter in comparison to the mean frequency (0.1s vs 0.5s for the mean frequency) which may explain this difference. 

The Shannon entropy showed an increase with increasing load for the aluminium sample. This increase is not seen in the corresponding increasing load test of the steel sample. It is interesting to note the large difference between the two sensors for both samples when under constant load. ” 

Round 2

Reviewer 1 Report

-

Author Response

Many thanks for giving a final review and finding our work fit for submission.

Reviewer 2 Report

The authors have addressed all my comments. After revision, I believe this paper is worth of publication.

Author Response

Reviewer’s comment

Authors’ response

Changes in the manuscript

(X) English language and style are fine/minor spell check required

We have carefully reviewed the manuscript in detail and have made numerous minor corrections to improve the quality of the English language, improve the clarity and consistency, and remove small typographical errors.

Please refer to the many changes throughout the revised manuscript.  There are too many changes to individually mention here.

The authors have addressed all my comments. After revision, I believe this paper is worth of publication.

Thank you once again for your insightful comments in the first round of revision.

No change.

Reviewer 3 Report

Authors improved their work. The novelty of the current work still should be

improved, such as  abstract.

Author Response

Reviewer’s comment

Authors’ response

Changes in the manuscript

(X) Extensive editing of English language and style required

We have carefully reviewed the manuscript in detail and have made numerous minor corrections to improve the quality of the English language, improve the clarity and consistency, and remove small typographical errors.

Please refer to the many changes throughout the revised manuscript.  There are too many changes to individually mention here.

Authors improved their work. The novelty of the current work still should be improved, such as abstract.

We have highlighted the novelty of this work in abstract, introduction and conclusions.

Additionally, Shannon Entropy and Centroid frequency analysis of single grit tests have now been included, and discussion and conclusions reworked to accommodate this additional analysis.

At line 13, in the abstract we have added the following:

“The results showed that AE parameters mean frequency, Centroid frequency and Shannon Entropy outperformed other frequency domain techniques by discriminating between the two sheet materials in scratch tests.”

Added to line 99:

“… to investigate the AE response of different materials when the applied load is insufficient to produce galling with Fourier analysis, mean frequency, centroid frequency and Shannon Entropy.

Figure 22 and 23 added on line 355 and line 358.

At line 344, the following text was added:

“Figure 22 shows that the Centroid Frequency for each of the SG tests is significantly different from each other. There is no notable trend for each of the materials. Figure 23 shows the Shannon Entropy for each of the tests. There is an increase in Shannon Entropy with time for the EN8 and CSMX4, which rapidly drops to 0 as the AE event ends, whereas the In-conel 718 shows a gradual increase and decline across the duration of the AE event. The magnitude of the MARM-002 is notably smaller than the other three materials.”

Paragraph on line 371 moved to line 422 and the following paragraph added at line 433:

“For the SG tests, the Centroid frequency and Shannon Entropy techniques again appear to discriminate between some materials. For Shannon Entropy a difference in EN8 and CSMX4 is hard to spot, however the Inconel 718 and MARM 002 are different in shape and magnitude. The Centroid frequency analysis reflects this, with the EN8 and CSMX4 appearing very similar to one another.”

L438 to L441 states: “The Mean Frequency, centroid frequency and Shannon entropy parameters showed a difference not reflected in Fourier techniques. Steel had higher mean and centroid frequency (600 kHz) than aluminium (500 kHz), attributed to a difference in hardness, as demonstrated by a difference in depth of cut.

Mean frequency performs as well as centroid frequency.”

Added the following to the conclusions at line 462:

“Centroid frequency and Shannon entropy can be used to discriminate between some materials.”
